# Kinetic sculpting of the seven stripes of the Drosophila *even-skipped* gene

**Augusto Berrocal[1†], Nicholas C Lammers[2†], Hernan G Garcia[1,2,3,4]\*, Michael B Eisen[1,2,4,5,6]\***

[1]Department of Molecular & Cell Biology, University of California at Berkeley, Berkeley, United States; [2]Biophysics Graduate Group, University of California at Berkeley, Berkeley, United States; [3]Department of Physics, University of California at Berkeley, Berkeley, United States; [4]Institute for Quantitative Biosciences-QB3, University of California at Berkeley, Berkeley, United States; [5]Department of Integrative Biology, University of California at Berkeley, Berkeley, United States; [6]Howard Hughes Medical Institute, University of California at Berkeley, Berkeley, United States

**Abstract** We used live imaging to visualize the transcriptional dynamics of the *Drosophila melanogaster even-skipped* gene at single-cell and high-temporal resolution as its seven stripe expression pattern forms, and developed tools to characterize and visualize how transcriptional bursting varies over time and space. We find that despite being created by the independent activity of five enhancers, *even-skipped* stripes are sculpted by the same kinetic phenomena: a coupled increase of burst frequency and amplitude. By tracking the position and activity of individual nuclei, we show that stripe movement is driven by the exchange of bursting nuclei from the posterior to anterior stripe flanks. Our work provides a conceptual, theoretical and computational framework for dissecting pattern formation in space and time, and reveals how the coordinated transcriptional activity of individual nuclei shapes complex developmental patterns.

**\*For correspondence:**
hggarcia@berkeley.edu (HGG);
mbeisen@gmail.com (MBE)

[†]These authors contributed equally to this work

## Introduction

The patterns of gene expression that choreograph animal development are formed dynamically by an interplay between processes – transcription, mRNA decay and degradation, diffusion, directional transport and the migration of cells and tissues – that vary in both space and time. However the spatial aspects of transcription have dominated the study of developmental gene expression, with the role of temporal processes in shaping patterns receiving comparably little attention (*Bothma and Levine, 2013*; *Garcia et al., 2020*).

Gene expression patterns are dynamic on many levels. They form, change and disappear over time, often as cells, tissues, and organs are forming and moving in the developing embryo (*Lawrence, 1992*). Furthermore the transcriptional process that creates these patterns is itself highly dynamic. The classical view of transcription as a switch or a tunable rheostat has been replaced in recent years by the recognition that mRNA synthesis occurs in bursts, with promoters switching stochastically between an ON state where polymerases are loaded and begin elongating, and an OFF state where few or no new transcripts are initiated (*Figure 1A*; *Zenklusen et al., 2008*; *Golding et al., 2005*; *Blake et al., 2006*; *Janicki et al., 2004*; *Chubb et al., 2006*; *Yunger et al., 2010*; *Raj et al., 2006*; *Lionnet et al., 2011*; *Muramoto et al., 2012*; *Little et al., 2013*; *Xu et al., 2015*; *Lenstra et al., 2015*; *Fukaya et al., 2016*; *Desponds et al., 2016*; *Hendy et al., 2017*; *Lammers et al., 2020*).

A slew of studies, from theoretical models (*Ko, 1991*; *Peccoud and Ycart, 1995*; *Sánchez and Kondev, 2008*; *Sanchez et al., 2011*; *Xu et al., 2016*; *Choubey et al., 2015*, *Choubey et al., 2018*;

*Shahrezaei and Swain, 2008*; *Kepler and Elston, 2001*; *Sasai and Wolynes, 2003*) to imaging-based analyses (*Xu et al., 2015*; *Fukaya et al., 2016*; *Senecal et al., 2014*; *Jones et al., 2014*; *Golding et al., 2005*; *Molina et al., 2013*; *Suter et al., 2011*; *So et al., 2011*; *Padovan-Merhar et al., 2015*; *Bartman et al., 2016*; *Hendy et al., 2017*; *Zoller et al., 2018*), have shown that overall rates of mRNA synthesis can be adjusted by controlling the bursting process. Changing the duration or bursts, the separation between bursts, or the rate at which polymerases are loaded during a burst (*Figure 1B*) will affect mRNA production, and modulating any or all of these parameters over space and time could, in principle, produce arbitrarily complex output patterns. However, it remains unclear how diverse the kinetic strategies employed by different regulatory sequences actually are, and what, if anything, constrains how these different kinetic parameters are used by evolution to shape patterns of expression.

In this paper, we set out to compare the ways in which different enhancers that drive similar types of spatiotemporal patterns during animal development deploy transcriptional bursting to produce their outputs by examining transcription at the single-cell level in living embryos. We use as our model the *Drosophila melanogaster even-skipped* (*eve*) gene whose seven stripes ring the embryo in the cellularizing blastoderm (nuclear cycle 14; nc14) in the hour preceding gastrulation (*Surkova et al., 2008*; *Fowlkes et al., 2008*; *Jiang et al., 2015*; *Ludwig et al., 2011*; *Macdonald et al., 1986*; *Frasch and Levine, 1987*).

The *eve* stripes are produced through the largely independent activity of five discrete enhancers (*Figure 1C*) that drive individual stripes (the stripes 1, 2, and 5 enhancers) or pairs of stripes (the stripe 3/7 and stripe 4/6 enhancers) (*Harding et al., 1989*; *Goto et al., 1989*; *Small et al., 1991*). These enhancers respond in different ways to canonical maternal factors Bicoid (Bcd) and Caudal (Cad), and gap genes Hunchback (Hb), Giant (Gt), Krüppel (Kr), Knirps (Kni) and Sloppy Paired 1 (Slp1), among others, balancing activating and repressive inputs to generate novel output patterns (*Mannervik, 2014*). For example, the *eve* stripe 2 enhancer is activated in the anterior by Bcd and Hb, and repressed by Gt and Kr, ultimately expressing in a stripe of nuclei that fall between the domains occupied by these two repressors (*Frasch and Levine, 1987*; *Small et al., 1992*).

Transcriptional bursting is widespread during *D. melanogaster* development (*Little et al., 2013*; *Xu et al., 2015*; *Bothma et al., 2014*; *Fukaya et al., 2016*; *Boettiger and Levine, 2013*; *Holloway and Spirov, 2017*; *Zoller et al., 2018*; *Paré et al., 2009*; *Lammers et al., 2020*). For example, *Bothma et al., 2014* utilized the MS2 system, which exploits the interaction between the phage MS2 coat protein (MCP) and a short RNA stem loop to fluorescently label nascent transcripts as they are being synthesized (*Bertrand et al., 1998*; *Garcia et al., 2013*; *Forrest and Gavis, 2003*), to directly visualize and quantify transcription from an *eve* stripe 2 transgene at single-nucleus resolution. They showed that the stripe is generated by bursts of transcriptional activity in the nuclei that form it, and that the aggregate pattern is highly dynamic, forming and dissipating rapidly during nc14.

Our objective in carrying out this work was twofold: first, to characterize the detailed dynamics of this classical and well-studied pattern as a means to reveal how multiple enhancers dictate potentially distinct bursting dynamics to shape a gene expression in the embryo, and second, to establish a rigorous systematic framework for analyzing such data, and conceptual paradigms for characterizing what we observe from this new type of experiment. Indeed, the advent of live imaging in the context of development calls for the establishment of a new language and new metrics for characterizing the formation of gene expression patterns in space and time.

We use a variety of new analyses to generate a kinetic fingerprint of *eve* transcription during stripe formation - a record of temporal and spatial variation in the bursting state of the promoters of ~3000 nuclei covering all seven stripes throughout nc14 - and to visualize different aspects of *eve* regulation. We find that all seven *eve* stripes are sculpted by the same regulatory strategies: the elimination of new bursts between stripes; the enhancement of bursting across stripes through a coupled increase in $k_{on}$ and $r$; and the refinement and movement of stripe positions by the addition of bursting nuclei along the anterior edge of the stripes and the loss of bursting along their posterior edge.

Thus, in this experiment and with our new set of analytical tools, we capture not only how single cell transcriptional activity encodes the formation of the stripes, but also how this activity is modulated in space and time in order to create and displace a complex pattern of gene activity across the embryo.

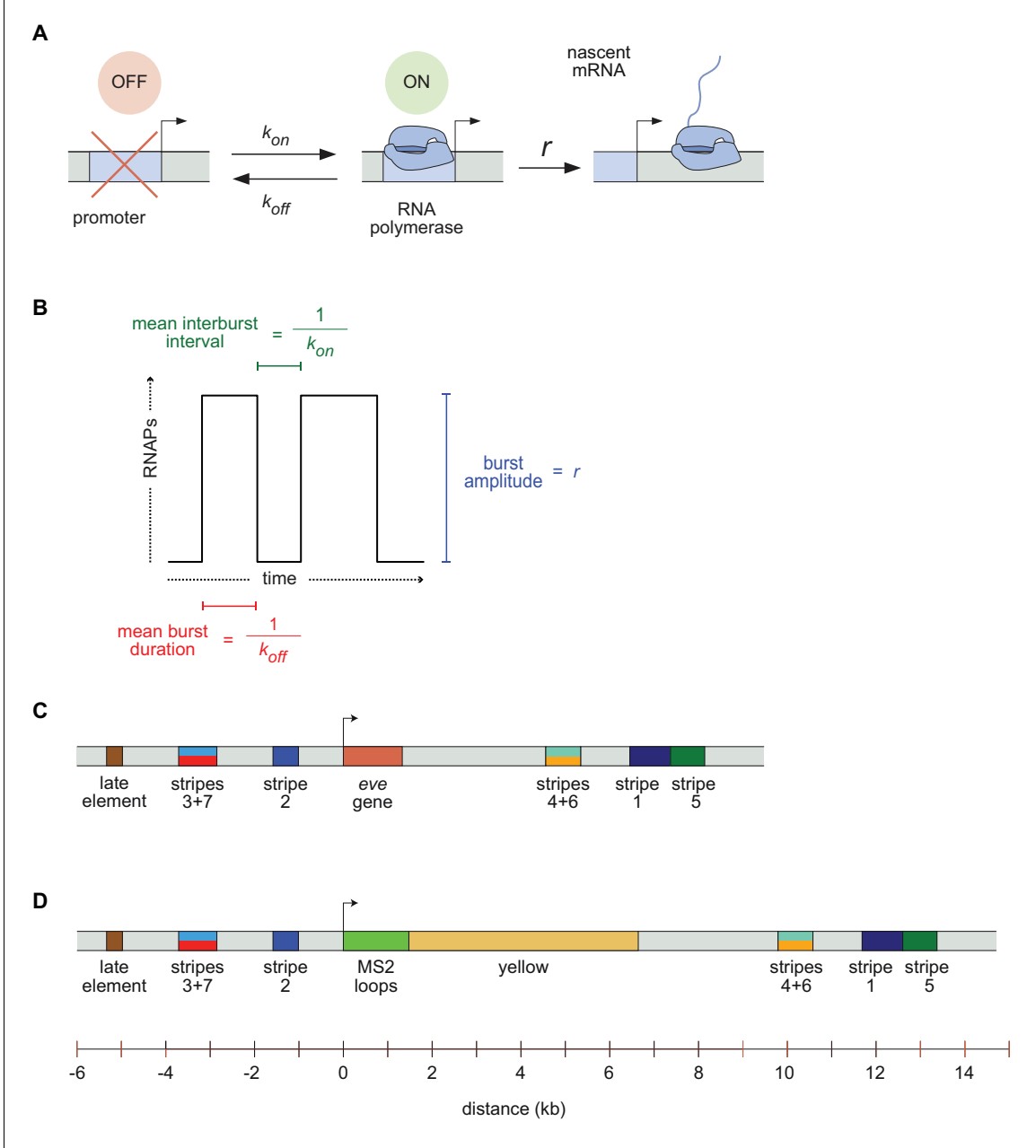

**Figure 1.** Visualizing live transcription from the seven stripes of *D. melanogaster even-skipped*. (A) Simple model of transcriptional bursting by promoter switching between ON and OFF states. (B) The promoter switching parameters define the burst duration, the duration between bursts, and amplitude. (C) Wild-type *eve* locus showing the five stripe enhancers (1, 2, 3+7, 4+6, 5) and the late enhancer element. Colors for individual stripes are used throughout figures. (D) Layout of the engineered *eve* BAC showing the locations of the MS2 stem loop array and *yellow* gene.

## Results

### Live imaging of eve expression

We used recombineering (*Warming et al., 2005*) to modify a bacterial artificial chromosome (BAC) (*Venken et al., 2006*) containing the *D. melanogaster eve* gene and all of its enhancers and regulatory elements (*Venken et al., 2009*), replacing the coding region with an array of MS2 stem loops followed by the *D. melanogaster yellow* (*y*) gene (*Figure 1D*; *Bothma et al., 2014*). The 4329 base pair *y* gene, which is longer than the endogenous *eve* transcript, is phenotypically neutral and provides a means to increase the number of RNA Polymerase II (Pol II) molecules loaded onto the gene

in order to amplify the signal (see Materials and methods for a discussion of how the structure of the reporter genes affects the fluorescence signal, analyses and inferences performed throughout this work). We inserted the engineered BAC into a targeted site on chromosome 3L using ΦC31 inte-grase-mediated recombination (*Fish et al., 2007*), and homozygosed the line, which showed no signs of adverse effects of the transgene.

We crossed males from this line with females carrying transgenes that express in embryos an RFP-labeled histone to visualize nuclei, and an MCP-GFP fusion that binds the MS2 RNA stem loops. The result is the direct visualization and quantification of nascent transcripts at the transgene locus as fluorescent puncta (*Garcia et al., 2013*). The temporal and spatial pattern of *eve* transgene transcription recapitulates the well-characterized dynamics of *eve* expression, most notably formation of the characteristic seven stripes in the late blastoderm (*Figure 2*; *Video 1*; *Surkova et al., 2008*; *Fowlkes et al., 2008*; *Jiang et al., 2015*; *Ludwig et al., 2011*). Further, as recently demonstrated in *Lammers et al., 2020* this BAC reporter construct quantitatively recapitulates the cytoplasmic *eve* mRNA pattern as measured by FISH (*Lammers et al., 2020*; *Luengo Hendriks et al., 2006*; *Lim et al., 2018*).

We used laser-scanning confocal microscopy to record, with high temporal resolution and high magnification, two color (MCP-GFP and histone-RFP) movies of embryos from before nc14 through gastrulation. We optimized our data collection strategy to sample multiple stripes (3 to 5) in each movie, to obtain high temporal resolution (one Z-stack, corresponding to each time point of our

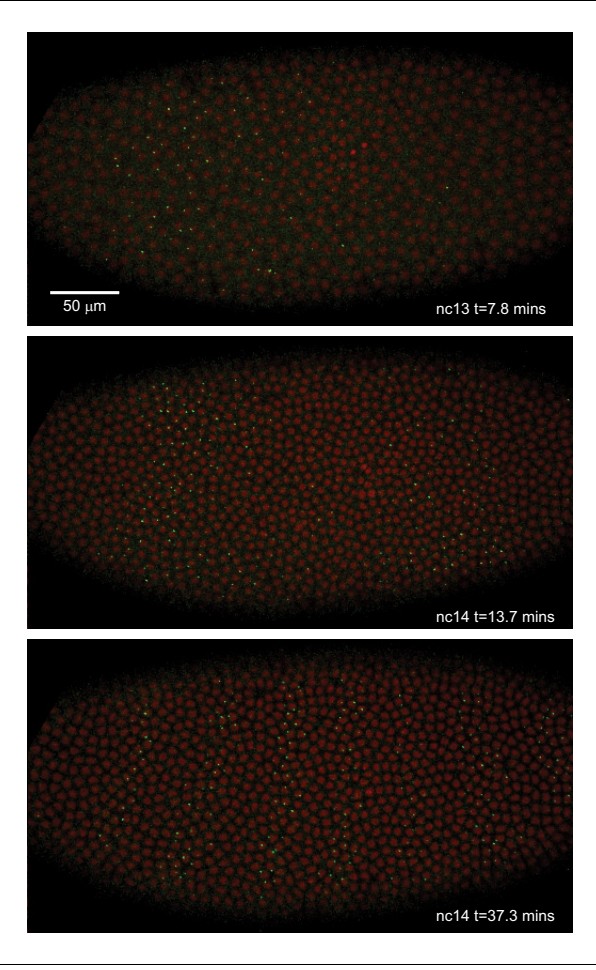

**Figure 2.** Live expression of *even-skipped*. Stills from maximum projection renderings of image stacks of an embryo spanning all seven stripes. This movie was collected with a 40x objective for illustration purposes only. Movies used for data analysis were collected at higher resolution as described in the text.

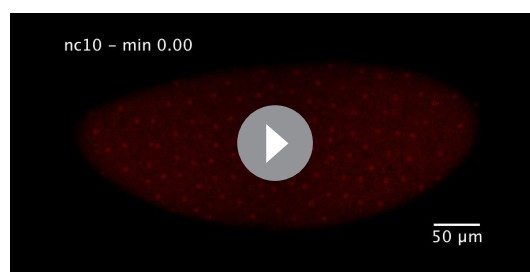

**Video 1.** Full expression pattern of *eve*-MS2 BAC. Maximum value projection of Z-stacks of an entire embryo carrying *eve*-MS2 BAC, MCP-GFP, and histone-RFP imaged with a 40x objective.
https://elifesciences.org/articles/61635#video1

**Video 2.** Individual dataset EVE_D1. Maximum value projection of Z-stacks of sections of embryos carrying *eve*-MS2 BAC, MCP-GFP, and histone-RFP imaged with a 63x objective, each capturing 3–5 stripes as described in *Table 1*.
https://elifesciences.org/articles/61635#video2

movies, every 16.8 s) and to have optimal signal-to-noise with minimal bleaching. In total, we collected 11 movies (*Videos 2–12*), with every stripe imaged at least five times (see *Table 1*).

We used a custom image processing pipeline (*Garcia et al., 2013*; *Lammers et al., 2020*) to identify nuclei, track fluorescent puncta and extract fluorescence intensities in each nucleus over time. The resulting data (*Supplementary file 1*) contains fluorescence traces from 2961 nuclei at an interpolated time interval of 20 s, representative examples of which are shown in *Figure 3A*.

We first sought to reexamine the previously characterized temporal dynamics of stripe formation (*Surkova et al., 2008*; *Fowlkes et al., 2008*; *Jiang et al., 2015*; *Ludwig et al., 2011*) using the increased temporal resolution (relative to earlier analyses of fixed embryos and of slowly maturing fluorescent protein reporters) of these data (*Figure 3B*). Early imaging studies described *eve* as being expressed broadly in nc13 and early nc14 embryos before refining sequentially into four, then seven stripes (*Macdonald et al., 1986*; *Frasch and Levine, 1987*). Subsequent work with improved labeling and imaging techniques (*Surkova et al., 2008*; *Fowlkes et al., 2008*) revealed an initial phase with broad domains in the anterior and posterior, followed by the formation of stripes from within these broad domains and, eventually, amplification of the stripe pattern.

During nc14, we first observe *eve* transcription beginning approximately five minutes after the onset of anaphase. The initial transcription covers a broad swath of the embryo, from around 25 to 75% egg-length, with the highest activity in two domains roughly centered in the anterior and posterior halves of the embryo, respectively. The greatest fluorescence signal during the first 25 min of nc14, when stripes are not yet fully formed, is in the most anterior region of *eve* transcription, in an area in which stripe one will eventually form.

Although the full seven stripe pattern is not fully formed until around 25 min, the three anterior-most stripes are already apparent as locally high areas of fluorescence intensity as early as 10 min. By 20 min stripes 1, 2, and 3 have clearly separated from background, stripes 4 and 6 appear to split off from a large posterior domain, and stripe 7 forms de novo in the posterior. Stripe 5 appears as a distinct stripe last, emerging in an area of low transcriptional activity left behind following the splitting of stripes 4 and 6. The stripes persist for the remainder of nc14, gradually increasing in fluorescence intensity until they reach a peak at around 35 min into nc14.

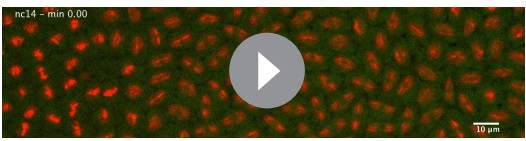

**Video 3.** Individual dataset EVE_D2. Maximum value projection of Z-stacks of sections of embryos carrying eve-MS2 BAC, MCP-GFP, and histone-RFP imaged with a 63x objective, each capturing 3–5 stripes as described in *Table 1*.
https://elifesciences.org/articles/61635#video3

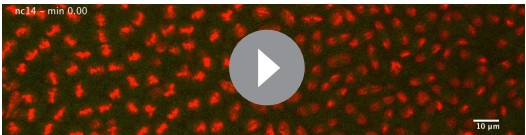

**Video 4.** Individual dataset EVE_D3. Maximum value projection of Z-stacks of sections of embryos carrying eve-MS2 BAC, MCP-GFP, and histone-RFP imaged with a 63x objective, each capturing 3–5 stripes as described in *Table 1*.
https://elifesciences.org/articles/61635#video4

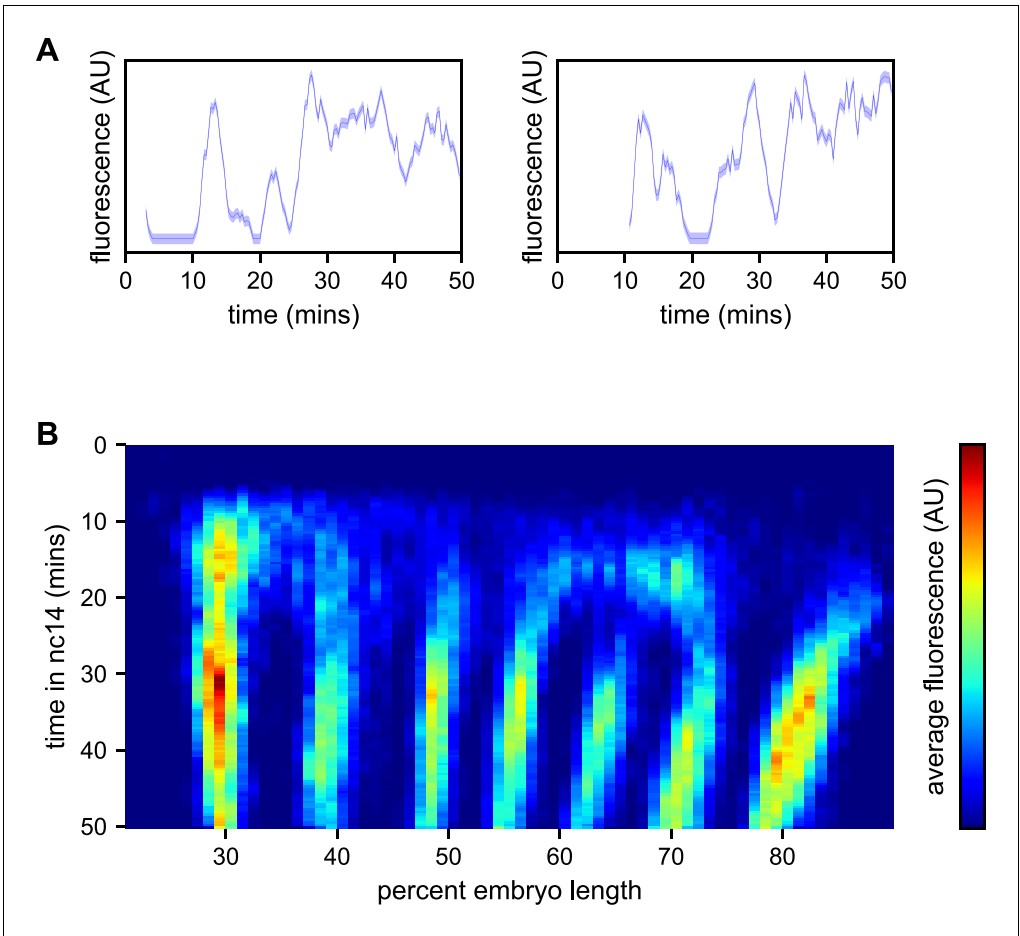

**Figure 3.** Spatiotemporal dynamics of *even-skipped* expression. (**A**) Fluorescence traces from two representative nuclei (particle ID = 1.0163 and 11.0448). (**B**) Average fluorescence over space and time showing stripe formation, modulation and movement. The time resolution along the y-axis is 20 s. The positions of nuclei along the x-axis were registered across movies based on the inferred position of stripe centers, and placed into bins of 1% embryo length, with the average fluorescence of all nuclei in each bin plotted. (A, shading corresponds to the error estimated based on the background fluorescence fluctuations as described in *Garcia et al., 2013*).

The positions of stripes 1–3 along the anterior-posterior (AP) axis are largely stable after they form, while stripes 4–6 show small anterior shifts. Stripe 7 makes a more dramatic movement towards the anterior, moving approximately 8% of egg-length, or around 40 μm from its initial location. The quantitative characterization of this stripe movement, the decoupling between stripes and nuclei, and the quantification of transcriptional bursting dynamics in each nucleus necessitated the development of a method, described below, to dynamically define the position of stripes throughout each movie.

## Modeling and inference of promoter state

As expected, the fluorescence traces from individual nuclei show clear hallmarks of transcriptional bursting, with apparent stochastic fluctuations between states with low and high fluorescence output (*Figure 3A*). Following previous work in the field (*Golding et al., 2005*; *Chubb et al., 2006*; *Zenklusen et al., 2008*; *Lionnet et al., 2011*; *Muramoto et al., 2012*; *Little et al., 2013*; *Xu et al., 2015*; *Lenstra et al., 2015*; *Fukaya et al., 2016*; *Desponds et al., 2016*; *Hendy et al., 2017*; *Zoller et al., 2018*; *Bothma et al., 2014*; *Paré et al., 2009*; *Lim et al., 2018*), we model bursting as a simple Markovian process in which a promoter switches stochastically between an OFF and an ON state with rates $k_{on}$ and $k_{off}$. When the promoter is in the ON state, we assume it loads polymerases continuously with a constant rate $r$ (*Figure 1A*).

**Table 1.** Summary of movies collected.

| Embryo ID | Duration | Stripes | Data movies | Promoter state movies |
|---|---|---|---|---|
| EVE_D1 | 255 frames 71.2 min | 1–4 | *Video 2* | *Video 13* |
| EVE_D2 | 254 frames 70.9 min | 3–7 | *Video 3* | *Video 14* |
| EVE_D3 | 235 frames 65.6 min | 3–6 | *Video 4* | *Video 15* |
| EVE_D4 | 246 frames 68.7 min | 3–7 | *Video 5* | *Video 16* |
| EVE_D5 | 210 frames 58.6 min | 4–7 | *Video 6* | *Video 17* |
| EVE_D6 | 196 frames 54.7 min | 4–7 | *Video 7* | *Video 18* |
| EVE_D7 | 208 frames 58.1 min | 3–7 | *Video 8* | *Video 19* |
| EVE_D8 | 232 frames 64.8 min | 1–3 | *Video 9* | *Video 20* |
| EVE_D9 | 322 frames 89.9 min | 1–4 | *Video 10* | *Video 21* |
| EVE_D10 | 267 frames 74.5 min | 1–3 | *Video 11* | *Video 22* |
| EVE_D11 | 307 frames 85.7 min | 1–4 | *Video 12* | *Video 23* |

In our implementation of the MS2 system, once a polymerase molecule transcribes the stem loops located at the 5' end of the gene, the MCP-GFP molecules bound to the stem loops produce a constant fluorescent signal at the locus that persists until this polymerase completes its traversal of the gene. Building off of the method presented in *Lammers et al., 2020*, we estimated this polymerase transit time as the displacement that gives the minimum value in the autocorrelation of the single frame differences in the fluorescent signal (see Materials and methods). The rationale for this approach was that every increase in signal due to polymerase loading at time $t$ should be accompanied by a corresponding decrease in signal at time $t+t_{elong}$ due to the completion of a transcriptional elongation cycle with a delay equal to the elongation time (*Coulon and Larson, 2016*; *Desponds et al., 2016*). We arrived at an estimate of 140 s (*Figure 4A*), consistent with a direct measurement of the rate of polymerase elongation of ~2700 bp/min from *Fukaya et al., 2017* and the length of the construct (6,563 bp).

We model the bursting process at each promoter in discrete time steps of $\Delta t$ = 20 s, set by the time resolution of our imaging. Under our model, in each time window a promoter is either OFF and not loading polymerases, or ON and loading polymerases at a fixed rate. A promoter that is in the ON state loads $\Delta t$ x $r$ polymerases, producing a single pulse of fluorescence proportional to $\Delta t$ x $r$ (with the proportionality factor determined by the fluorescence of GFP and the fraction of MS2 loops bound by MCP-GFP). This pulse lasts at the locus for 140 s, at which point all polymerase molecules loaded during the original time window have terminated transcribing (*Figure 4B*). Since we do not calibrate the fluorescence signal to the number of polymerase molecules for this construct, in practice we fold the proportionality factor into $r$ altering its units from polymerases loaded per unit time to fluorescence signal produced per unit time. Since many transcriptional bursts last for longer than 20 s, the fluorescence output of a single burst is a sum of the pulses generated during each time window.

In the embryos we imaged here, the MS2 BAC is heterozygous, contributed only by the father, while the mother contributes the MCP-GFP. However, DNA replication occurs within an average of 10 min for loci in nc14 (*McKnight and Miller, 1977*; *Rabinowitz, 1941*; *Shermoen et al., 2010*), meaning that there are actually two sister chromatids with the MS2-containing transgene in every nucleus. Because of sister chromatid cohesion, we cannot, in general, discriminate both copies (*Fung et al., 1998*; *Wilkie et al., 1999*; *Little et al., 2013*). Thus we model the locus as having two

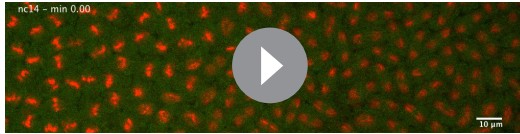

**Video 5.** Individual dataset EVE_D3. Maximum value projection of Z-stacks of sections of embryos carrying eve-MS2 BAC, MCP-GFP, and histone-RFP imaged with a 63x objective, each capturing 3–5 stripes as described in *Table 1*.
https://elifesciences.org/articles/61635#video5

**Video 6.** Individual dataset EVE_D5. Maximum value projection of Z-stacks of sections of embryos carrying eve-MS2 BAC, MCP-GFP, and histone-RFP imaged with a 63x objective, each capturing 3–5 stripes as described in *Table 1*.
https://elifesciences.org/articles/61635#video6

promoters, which implies that the system can be in one of three distinct states: OFF, one promoter ON, and two promoters ON. For ease of exposition, however, we will frame our discussion in terms of an effective 2-state model in which the locus is ON so long as *at least* one promoter is in the ON state. As it is still unclear how the sister chromatids influence each other's transcription (*McKnight and Miller, 1977*; *Lammers et al., 2020*; *Zoller et al., 2018*), we make no prior assumptions about the nature or degree of correlations between sister chromatids (though, for illustration purposes, in *Figure 4B* we entertain the case where sister chromatids transcribe independently).

As Illustrated in *Figure 4C* for the case of independent sister chromatids, if we know the state of the promoter over time, we can reconstruct its expected fluorescence output by summing 140 s pulses beginning at each point where the promoter is ON and having height $r$ if one promoter is ON or height $2 \times r$ if two promoters are ON. Traces modeled from hypothetical promoter state sequences (*Figure 4C*) have the features of the observed fluorescence signal: linear increases in intensity (corresponding to periods when the promoter is ON); plateaus (corresponding to periods when transcriptional initiation is matched with previously initiated polymerases completing their transit of the gene); and linear signal decays (corresponding to periods when the promoter is OFF but previously initiated polymerases are still transiting the gene) (*Bothma et al., 2014*; *Garcia et al., 2013*).

However, when given a fluorescence trace, it is not trivial to infer the promoter state sequence that generated it, owing to the time convolution between promoter state and fluorescence output. To solve this problem, we developed a compound-state hidden Markov Model (cpHMM, described in *Lammers et al., 2020*) that estimates global parameters $k_{on}$, $k_{off}$, and $r$ for a set of traces, and allows us to identify the maximum-likelihood promoter state sequence under these parameters for every trace via the Viterbi algorithm.

The cpHMM thus accomplishes two aims central to treating these data in a more rigorous and biologically meaningful manner. First, it allows us to describe the bursting behavior of any set of nuclei in quantitative terms. Across all seven stripes, the model infers approximate $k_{on}$ $k_{off}$ values of 0.60 events per minute and an $r$ of 67 AU per minute. And second, by providing a means to fit a sequence of ON and OFF states to the data from each nucleus, the cpHMM allows us to shift the focus in the analysis of individual traces from fluorescence, which only indirectly reflects the temporal behavior of a promoter, to the instantaneous promoter state (*Figure 4D–F*; see also

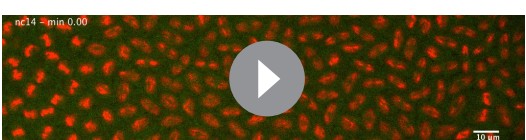

**Video 7.** Individual dataset EVE_D6. Maximum value projection of Z-stacks of sections of embryos carrying eve-MS2 BAC, MCP-GFP, and histone-RFP imaged with a 63x objective, each capturing 3–5 stripes as described in *Table 1*.
https://elifesciences.org/articles/61635#video7

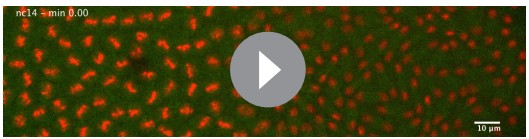

**Video 8.** Individual dataset EVE_D7. Maximum value projection of Z-stacks of sections of embryos carrying eve-MS2 BAC, MCP-GFP, and histone-RFP imaged with a 63x objective, each capturing 3–5 stripes as described in *Table 1*.
https://elifesciences.org/articles/61635#video8

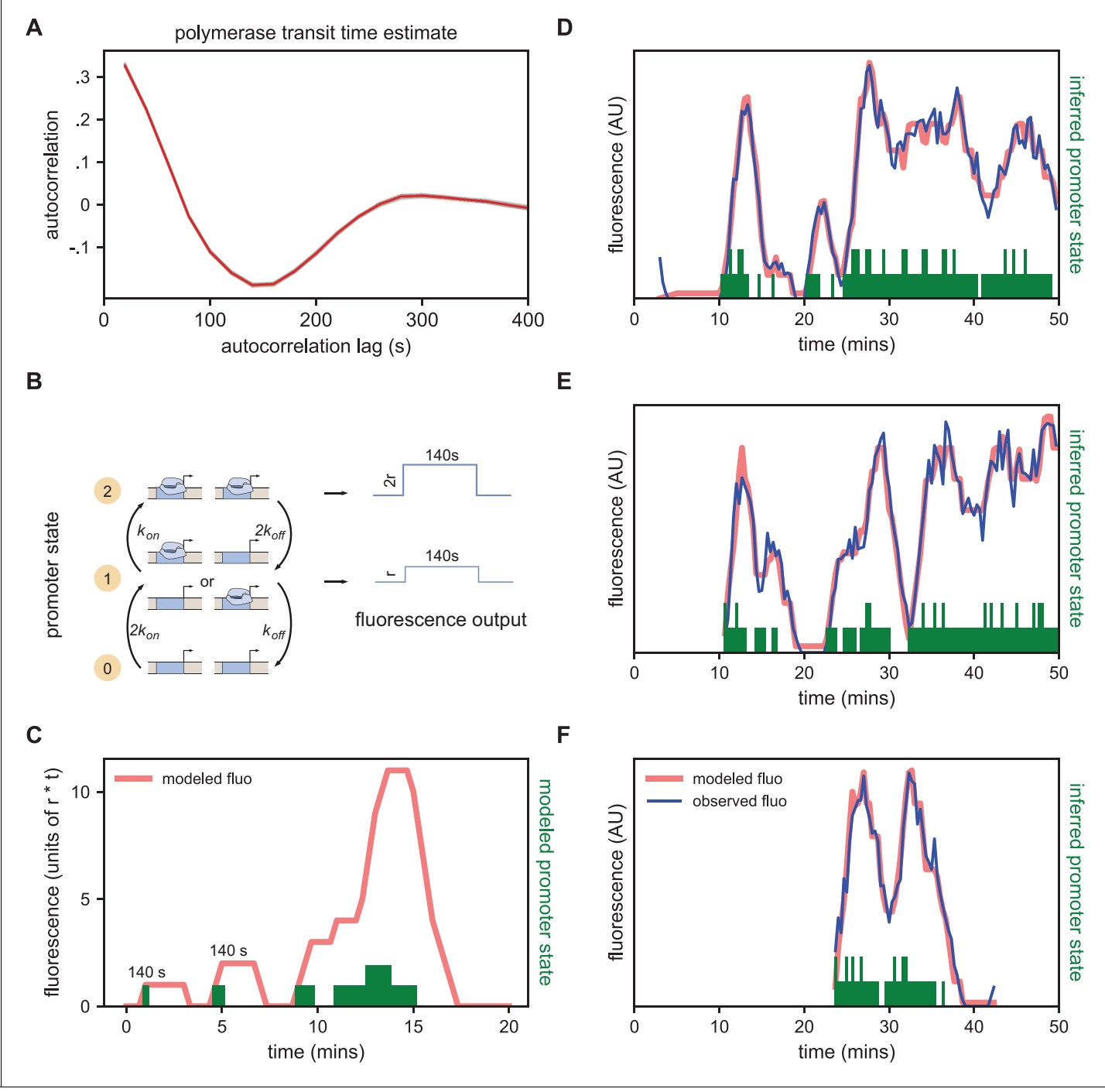

**Figure 4.** Modeling bursting in individual nuclei. (**A**) A key parameter in relating fluorescence output to the bursting state of a promoter is the time it takes for a polymerase to transit the gene, which we determined as approximately 140 s by examining the autocorrelation (red line) of the change in fluorescence. Gray lines show 100 bootstraps over randomly selected sets of 80% of nuclei; note they almost perfectly overlap the red line. (**B**) Three state model accounting for post-replication presence of sister chromatids. When either promoter is ON for a short time period Δt, it loads polymerases at a constant rate contributing a pulse of polymerase that persists for 140 s. For the inference results presented in subsequent sections, states 1 and 2 are subsumed into a single effective ON state to give rise to an effective two-state model. The hypothetical rates depicted here assume that each promoter bursts independently, an assumption that we relax for the burst parameter inference. (**C**) Simplified example of the expected observed fluorescence (red line) produced from a hypothetical promoter state sequence. The fluorescence is the sum of the fluorescence pulses produced when one or both promoters are ON (given by the height of the green bars). (**D–F**) Representative fluorescence traces from individual nuclei (blue lines), inferred bursting pattern (green bars) and fluorescence imputed by cpHMM (red line) for particles 1.0163 (**D**), 11.0448 (**E**) and 5.0231 (**F**).

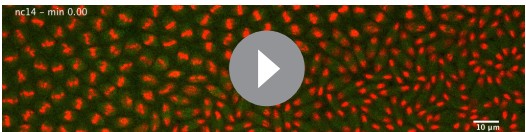

**Video 9.** Individual dataset EVE_D8. Maximum value projection of Z-stacks of sections of embryos carrying eve-MS2 BAC, MCP-GFP, and histone-RFP imaged with a 63x objective, each capturing 3–5 stripes as described in *Table 1*.
https://elifesciences.org/articles/61635#video9

**Video 10.** Individual dataset EVE_D9. Maximum value projection of Z-stacks of sections of embryos carrying eve-MS2 BAC, MCP-GFP, and histone-RFP imaged with a 63x objective, each capturing 3–5 stripes as described in *Table 1*.
https://elifesciences.org/articles/61635#video10

*Supplementary file 1* which the inferred promoter state for each nucleus at every time point and the corresponding modeled fluorescence intensity, and *Videos 13–23*).

## Dynamic determination of stripe positions

Before analyzing the data further we had to solve two practical problems. To compare the kinetic behavior of individual stripes, we had to determine which nuclei were in each stripe at every time point, a process complicated by the movement of stripes relative to both the embryo and nuclei. Further, to analyze the data *in toto*, we also had to register the 11 movies relative to each other and to the embryo.

To address these challenges, we used a Gaussian mixture model to cluster bursting nuclei in each movie in a series of overlapping six-minute time windows based on their x and y positions in the image (*Figure 5A*). This clustering reliably separates nuclei into individual stripes. We next determined the orientation of each stripe to the AP and imaging axes by fitting a line to coordinates of all nuclei assigned to that stripe in each movie (*Figure 5B*). We fit a line with this slope to bursting nuclei from each time window (*Figure 5C and D*), and use these fits to generate a linear model of the position of each stripe in each image over time, which we use to reorient the stripe so that it is perpendicular to the image x-axis (*Figure 5E*).

We next use the known coordinates of the anterior and posterior poles of the embryo to convert the image x-axis to AP position, and register the examples of each stripe from different movies by setting the AP position of the center of each stripe at 35 min in nc14 to the mean AP position of all examples of that stripe at 35 min, adjusting the position of the stripe at other time points by the same correction (*Figure 5F*). As the stripes are not all present until after 25 min in nc14, we assign and register nuclei before that point based on the stripe position at 25 min. The stripe assignment is invariant over bootstrapping of movies, and the standard deviation of the AP displacement over bootstrapping of movies is 0.16% of embryo length.

Collectively these data represent an easy way to visualize and interpret kinetic fingerprint of stripe formation: a record of every transcriptional burst that occurred during the formation of *eve* stripes in these embryos (*Figure 6*; *Video 24*).

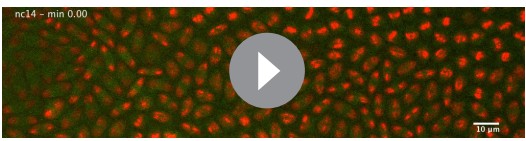

**Video 11.** Individual dataset EVE_D10. Maximum value projection of Z-stacks of sections of embryos carrying eve-MS2 BAC, MCP-GFP, and histone-RFP imaged with a 63x objective, each capturing 3–5 stripes as described in *Table 1*.
https://elifesciences.org/articles/61635#video11

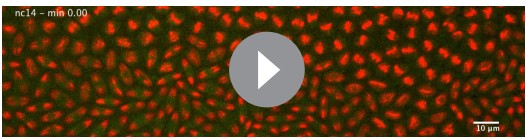

**Video 12.** Individual dataset EVE_D11. Maximum value projection of Z-stacks of sections of embryos carrying eve-MS2 BAC, MCP-GFP, and histone-RFP imaged with a 63x objective, each capturing 3–5 stripes as described in *Table 1*.
https://elifesciences.org/articles/61635#video12

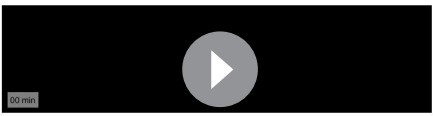

**Video 13.** Promoter state movie for dataset EVE_D1. Animation of pseudo-cells (resulting from a Voronoi tessellation based on the position of nuclei) where cells are colored based on their stripe, with intensity proportional to the measured *eve* MS2 fluorescence of the nucleus at the given time, and promoters in the ON and OFF states represented with light and dark gray pseudo-cell outlines, respectively.
https://elifesciences.org/articles/61635#video13

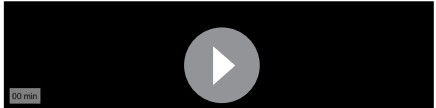

**Video 14.** Promoter state movie for dataset EVE_D2. Animation of pseudo-cells (resulting from a Voronoi tessellation based on the position of nuclei) where cells are colored based on their stripe, with intensity proportional to the measured eve MS2 fluorescence of the nucleus at the given time, and promoters in the ON and OFF states represented with light and dark gray pseudo-cell outlines, respectively.
https://elifesciences.org/articles/61635#video14

## Bursting dynamics of individual nuclei

We used the output of the cpHMM and registration process to examine the locations of transcriptional bursts along the AP axis and over time (*Figure 6*). The most striking feature is the almost complete lack of observable transcriptional bursts in the regions between stripes from 25 min into nc14, with the exception of the 5–6 interstripe which is discussed below (note that this is not an artifact of the movie alignment and orientation process, as this effect is seen clearly in individual movies). We took advantage of the fact that we were tracking bursts in individual nuclei in order to analyze the relationship between this absence of bursting in interstripe regions and the single-nucleus bursting behavior within stripes.

Stripes are defined by sharp spatial boundaries, with the transition between the low-bursting (quiescent) state and the frequently bursting (active) state occurring from one column of nuclei to the next (*Figure 6*), consistent with the classical descriptions of *eve* stripe patterns (*Small et al., 1991*; *Frasch and Levine, 1987*; *Fujioka et al., 1999*; *Small et al., 1992*; *Warrior and Levine, 1990*; *Clyde et al., 2003*). They also have sharp temporal boundaries: all of the interstripe regions, save that between stripes 6 and 7, form in regions where there was appreciable bursting early in nc14 that disappears at around 25 min into the nuclear cycle (*Figure 6*).

To better understand how the low-bursting state in interstripes is established, we looked at the bursting history of the nuclei in these regions (*Figure 7*). The first feature we noticed was that most of the nuclei that ultimately form the interstripe were never detected to burst at any point in nc14 (*Figure 7A,B*). With the exception of the 5–6 interstripe, these never-ON nuclei effectively form the boundaries between stripes, as essentially every nucleus within each stripe bursts at some point during nc14 (*Figure 7A,B*).

**Video 15.** Promoter state movie for dataset EVE_D3. Animation of pseudo-cells (resulting from a Voronoi tessellation based on the position of nuclei) where cells are colored based on their stripe, with intensity proportional to the measured eve MS2 fluorescence of the nucleus at the given time, and promoters in the ON and OFF states represented with light and dark gray pseudo-cell outlines, respectively.
https://elifesciences.org/articles/61635#video15

**Video 16.** Promoter state movie for dataset EVE_D4. Animation of pseudo-cells (resulting from a Voronoi tessellation based on the position of nuclei) where cells are colored based on their stripe, with intensity proportional to the measured eve MS2 fluorescence of the nucleus at the given time, and promoters in the ON and OFF states represented with light and dark gray pseudo-cell outlines, respectively.
https://elifesciences.org/articles/61635#video16

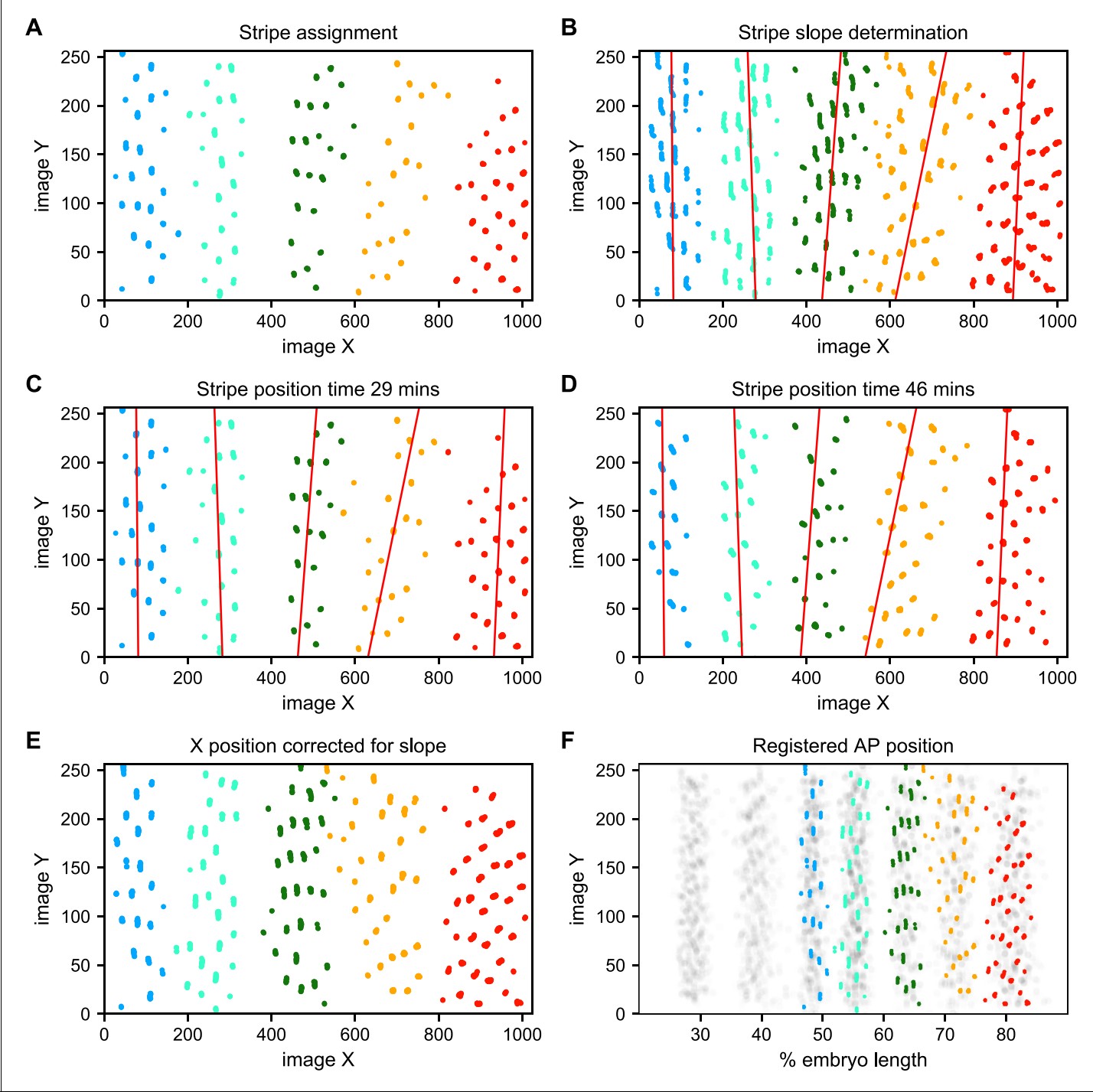

**Figure 5.** Stripe assignment and alignment. (**A**) We preliminarily assign bursting nuclei to stripes by applying a Gaussian mixture model to each movie independently in overlapping six-minute time windows with the number of Gaussians equal to the number of stripes captured in that movie. An example is shown here from **Video 2**. (**B,C,D**) We next determine the orientation of each stripe to the imaging axes by fitting a line to coordinates of all nuclei from t > 25 min assigned to that stripe in each movie and time window. (**E**) We use these fits to generate a linear model of the position of each stripe in each image over time, which makes it possible to reorient the stripe so that it is perpendicular to the image x-axis. (**F**) The known coordinates of the anterior and posterior poles of the embryo are used to convert the image x-axis to AP position and register the stripes from different movies to each other, as shown here for nuclei from **Video 2** colored by stripe and nuclei corresponding to all other movies drawn in gray.

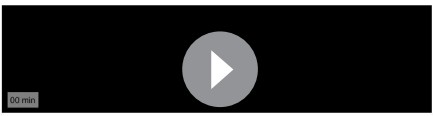

**Video 17.** Promoter state movie for dataset EVE_D5. Animation of pseudo-cells (resulting from a Voronoi tessellation based on the position of nuclei) where cells are colored based on their stripe, with intensity proportional to the measured eve MS2 fluorescence of the nucleus at the given time, and promoters in the ON and OFF states represented with light and dark gray pseudo-cell outlines, respectively.
https://elifesciences.org/articles/61635#video17

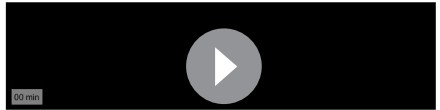

**Video 18.** Promoter state movie for dataset EVE_D6. Animation of pseudo-cells (resulting from a Voronoi tessellation based on the position of nuclei) where cells are colored based on their stripe, with intensity proportional to the measured eve MS2 fluorescence of the nucleus at the given time, and promoters in the ON and OFF states represented with light and dark gray pseudo-cell outlines, respectively.
https://elifesciences.org/articles/61635#video18

The contrast in bursting history between stripes and interstripes is less pronounced in the posterior, where there are fewer such never-ON nuclei in the interstripe region (*Figure 7B*, notice the lower density of red single-nuclei tracks corresponding to never-ON nuclei). In order to reveal the source of this reduced number of never-ON nuclei in posterior interstripes, we analyzed their bursting history. *Figure 7C* shows the AP positions of the nuclei in one movie covering stripe seven as a function of time, with the period in which they are part of the stripe highlighted. Although the stripe is clearly present throughout this period, no nuclei remain a part of the stripe for the entirety of this 25 min period. As time progresses, nuclei at the posterior edge of stripe seven shift from an active state, in which the promoter stochastically alternates between the ON and OFF transcriptional states, to a quiescent state in which we observe no appreciable bursting. In contrast, nuclei just off the anterior edge of the stripe switch from a quiescent to an active state at roughly the same rate. This leads to a net overall anterior movement of the stripe, akin to treadmilling, at a velocity of approximately one percent of embryo length every three minutes.

Consistent with *Lim et al., 2018*, the other stripes exhibit smaller and varied anterior shifts (*Figure 7—figure supplement 1*), but in every case the shift is associated with a similar coupled gain of active nuclei along the anterior edge and loss along the posterior edge. This effect is most clearly seen in *Figure 7D*, which shows, for each time point where a nucleus initiates a new burst, the difference in activity (defined as the difference between the fraction of the time the nucleus is in the ON state in the subsequent 10 min minus the fraction of the time the nucleus is in the ON state in the preceding 10 min). For all seven stripes there is a clear spatial pattern, with nuclei along the anterior edge of the stripe entering a bursting state and nuclei along the posterior edge becoming quiescent, indicating a movement of stripes relative to nuclei. Hence, stripe movement is associated with

**Video 19.** Promoter state movie for dataset EVE_D7. Animation of pseudo-cells (resulting from a Voronoi tessellation based on the position of nuclei) where cells are colored based on their stripe, with intensity proportional to the measured eve MS2 fluorescence of the nucleus at the given time, and promoters in the ON and OFF states represented with light and dark gray pseudo-cell outlines, respectively.
https://elifesciences.org/articles/61635#video19

**Video 20.** Promoter state movie for dataset EVE_D8. Animation of pseudo-cells (resulting from a Voronoi tessellation based on the position of nuclei) where cells are colored based on their stripe, with intensity proportional to the measured eve MS2 fluorescence of the nucleus at the given time, and promoters in the ON and OFF states represented with light and dark gray pseudo-cell outlines, respectively.
https://elifesciences.org/articles/61635#video20

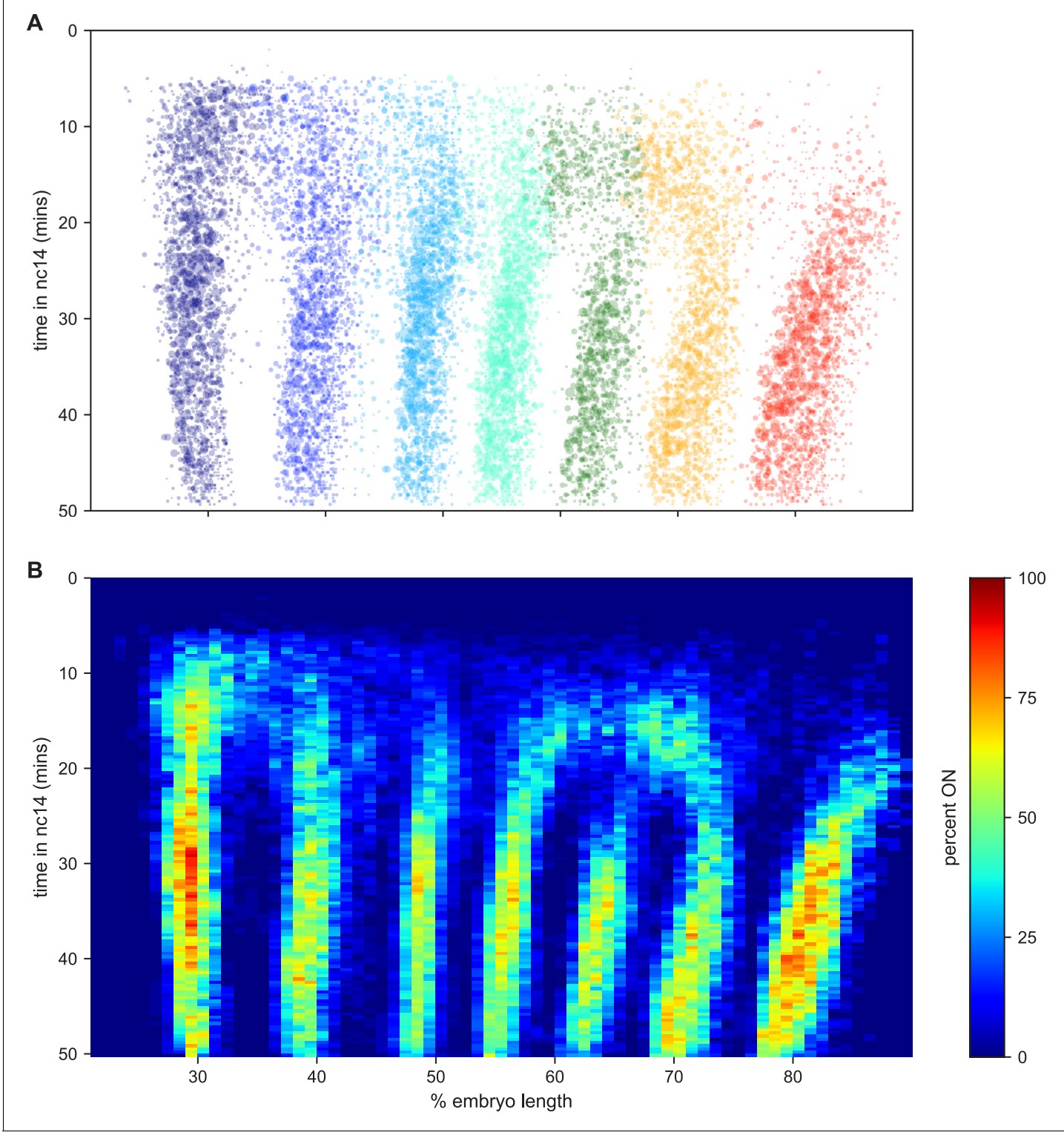

**Figure 6.** The kinetic fingerprint of *even-skipped* stripe formation. (**A**) Inferred location of every transcriptional burst in all 11 movies as a function of time and where along the anterior-posterior axis (plotted as fraction of embryo length) each burst occurred. The size of the dot represents the duration of the burst. Collectively the data create a kinetic fingerprint of *eve* stripe formation. (**B**) Instantaneous fraction of nuclei in the transcriptionally active ON state as a function of time and position along the embryo.

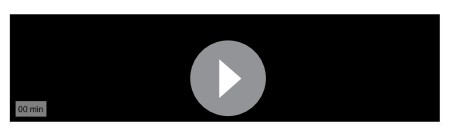

**Video 21.** Promoter state movie for dataset EVE_D9. Animation of pseudo-cells (resulting from a Voronoi tessellation based on the position of nuclei) where cells are colored based on their stripe, with intensity proportional to the measured eve MS2 fluorescence of the nucleus at the given time, and promoters in the ON and OFF states represented with light and dark gray pseudo-cell outlines, respectively.
https://elifesciences.org/articles/61635#video21

**Video 22.** Promoter state movie for dataset EVE_D10. Animation of pseudo-cells (resulting from a Voronoi tessellation based on the position of nuclei) where cells are colored based on their stripe, with intensity proportional to the measured eve MS2 fluorescence of the nucleus at the given time, and promoters in the ON and OFF states represented with light and dark gray pseudo-cell outlines, respectively.
https://elifesciences.org/articles/61635#video22

the dynamic switching of nuclei between active and quiescent states, and not just with the movement of nuclei themselves.

## All seven *eve* stripes are created by the same regulation of bursting kinetics

We next turned to the questions of how the spatial pattern of nuclear transcriptional activity described above is produced by regulating bursting kinetics, and whether this regulation differs among the seven *eve* stripes. In principle, any pattern of transcriptional activity could be achieved by modulating the duration, separation and/or amplitude of bursts across space and time. For example, a stripe could be created by varying burst separation along the anterior-posterior axis, with nuclei in the stripe center having lower burst separation, and those outside the stripe having long periods without bursts, or no bursts at all. Alternatively, the same stripe could be created with uniform burst separation across nuclei, but elevated burst duration or amplitude within the stripe, or by modulating multiple parameters simultaneously.

Ideally, we would like to have a measure of the bursting parameters governing the behavior of every nucleus. However, individual MS2 traces have too few time points to allow for accurate cpHMM inference of burst parameters at the single trace level. We therefore used the cpHMM to infer $k_{on}$, $k_{off}$, and $r$ for groups of nuclei binned on their mean fluorescence output and stripe. The logic of the fluorescence binning was that, given that $\langle fluorescence \rangle \propto r \frac{k_{on}}{k_{on}+k_{off}}$ (**Lammers et al., 2020**), nuclei that have similar $k_{on}$, $k_{off}$, and $r$ will have similar fluorescence outputs. Our inference shows that $k_{on}$ is very strongly regulated as a

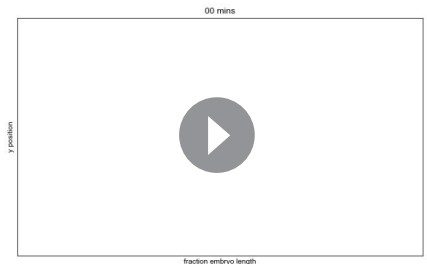

**Video 23.** Promoter state movie for dataset EVE_D11. Animation of pseudo-cells (resulting from a Voronoi tessellation based on the position of nuclei) where cells are colored based on their stripe, with intensity proportional to the measured eve MS2 fluorescence of the nucleus at the given time, and promoters in the ON and OFF states represented with light and dark gray pseudo-cell outlines, respectively.
https://elifesciences.org/articles/61635#video23

**Video 24.** Kinetic fingerprint of *eve* stripe formation. Nuclei are graphed at every time point at its registered AP (x-axis) and image y (y-axis) position when the cpHMM inferred that one copy (small circles) or two copies (large circles) of the promoter was in the ON state (see **Figure 4B**).
https://elifesciences.org/articles/61635#video24

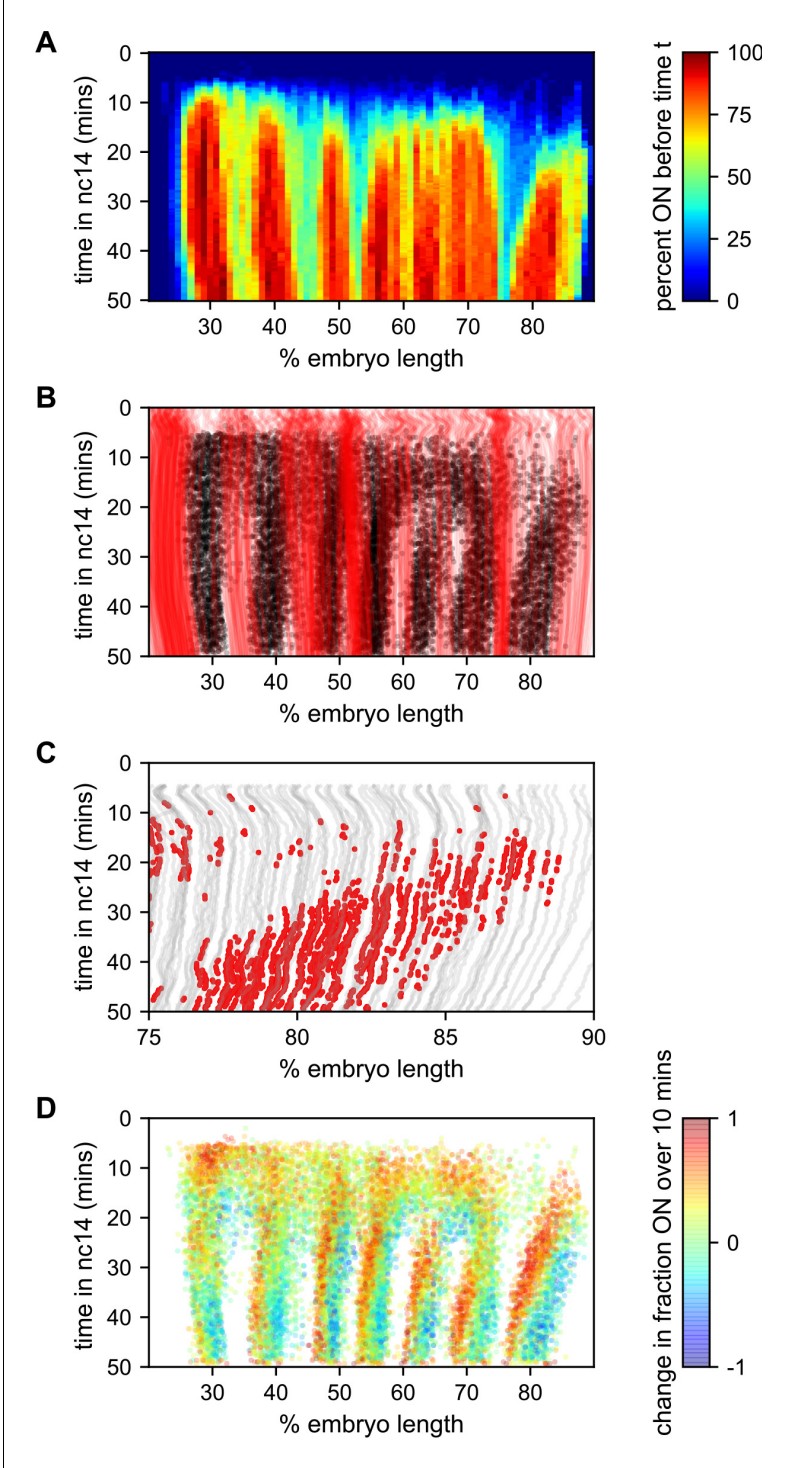

**Figure 7.** Stripe formation and movement. (**A**) Fraction of nuclei bursting before time *t* as a function of position along the embryo. (**B**) Locations of new bursts (black dots) in space and time along with spatiotemporal traces of nuclei that are in the OFF state throughout nc14 (red lines). (**C**) Traces of nuclei positions over time (gray lines) from stripe 7 region of movie EVE_D6 with timepoints where new bursts initiated colored red to illustrate stripe movement relative to nuclei. (**D**) Difference in transcriptional activity as defined as the difference between the fraction of the time each nucleus is in the ON state in the subsequent 10 min minus the fraction of time the nucleus is in the ON state in the preceding 10 min. Positive values represent a nucleus turning on or increasing activity, while blue values indicate a nucleus turning off or decreasing activity.

*Figure 7 continued on next page*

*Figure 7 continued*

The online version of this article includes the following figure supplement(s) for figure 7:

**Figure supplement 1.** Stripe movement is dominated by the movement of transcriptional activity.

function of average fluorescence in a consistent manner across stripes (*Figure 8A*). In contrast, only a weak drop in $k_{off}$ is observed (*Figure 8B*). Finally, $r$ also featured a strong upregulation as a function of average fluorescence across stripes (*Figure 8C*).

As shown in *Figure 8D*, each stripe contains nuclei with a relatively wide range of average fluorescence values. In order to reveal the bursting parameters across the AP axis for each stripe, we averaged the single-cell bursting parameters determined in each stripe (*Figure 8A–C*) weighted by the relative number of nuclei in each fluorescence bin present at each position along the AP axis (*Figure 8D*). We find that the variation in bursting parameters observed as a function of average fluorescence largely echoes the modulation of fluorescence in space (*Figure 8E*). Specifically, while there is a subtle downregulation of $k_{off}$ within stripes, $k_{on}$ and $r$ are significantly upregulated in the center of each stripe.

Thus, not only do the five *eve* enhancers employ a common regulatory strategy for modulating the fluorescence output of nuclei to create a stripe, decreasing burst separation and increasing burst amplitude with a constant burst duration, the precise quantitative relationship among these bursting parameters is maintained across a wide range of molecular inputs and fluorescence outputs.

## Discussion

The most remarkable aspect of *eve* regulation is that what appears to be a regular, repeating pattern of nearly identical stripes is created by the largely independent activity of five separate enhancers responding to different combinations of activators and repressors (*Fujioka et al., 1999*; *Fujioka et al., 1995*; *Small et al., 1991*; *Arnosti et al., 1996*; *Small et al., 1992*). We have now shown that the connection between the stripe enhancers is more than just that they produce the same kind of pattern: they realize these patterns through the same control of transcriptional bursting.

Although, in principle, complex patterns of transcription could be generated by the independent regulation of $k_{on}$, $k_{off}$ or $r$, many of the key features of *eve* stripe regulation we observe here involve the modulation of $k_{on}$ and $r$ in concert. The most straightforward explanation for this shared mode of bursting control is that there is a single molecular pathway via which *eve* transcriptional bursting is regulated, with enhancers essentially having access to only a single tunable parameter. Whether this parameter is determined by the gene through, for example, the promoter sequence, or whether this single molecular pathway reflects some broad common property of gene regulation, such as constraints on the general transcriptional machinery, remains an open question. The limited data on bursting control available for other genes in the fly (*Falo-Sanjuan et al., 2019*; *Fukaya et al., 2016*; *Zoller et al., 2018*) suggests that control mechanisms are not ubiquitously the same and that they might be unique to different classes of genes.

An alternative explanation for the observed commonalities in the control of bursting is that there is a functional reason to use this strategy. Namely, that this is not the result of a common molecular mechanism, but rather of common selective pressures acting on the five enhancers independently. The particular bursting control strategy uncovered here might, for example, be more robust to fluctuations in transcription factor concentrations or temperature, or provide more precise spatiotemporal gene expression control (*Shelansky and Boeger, 2020*; *Grah et al., 2020*). New experiments and theoretical work will be necessary in order to uncover the specific molecular pathways by which bursting is controlled and to understand the functional consequences of different bursting strategies that create the same mRNA levels.

In addition to this modulation of bursting, the fraction of nuclei that engage in transcription at any point in the nuclear cycle is higher in stripe centers than in interstripes. This regulation of the fraction of active nuclei, also seen in other genes (*Garcia et al., 2013*), seems to reside outside of the bursting framework. Such regulation, as well as the spatial modulation of the window of time over which bursting ensues, suggests the presence of multiple and overlapping modes of regulation

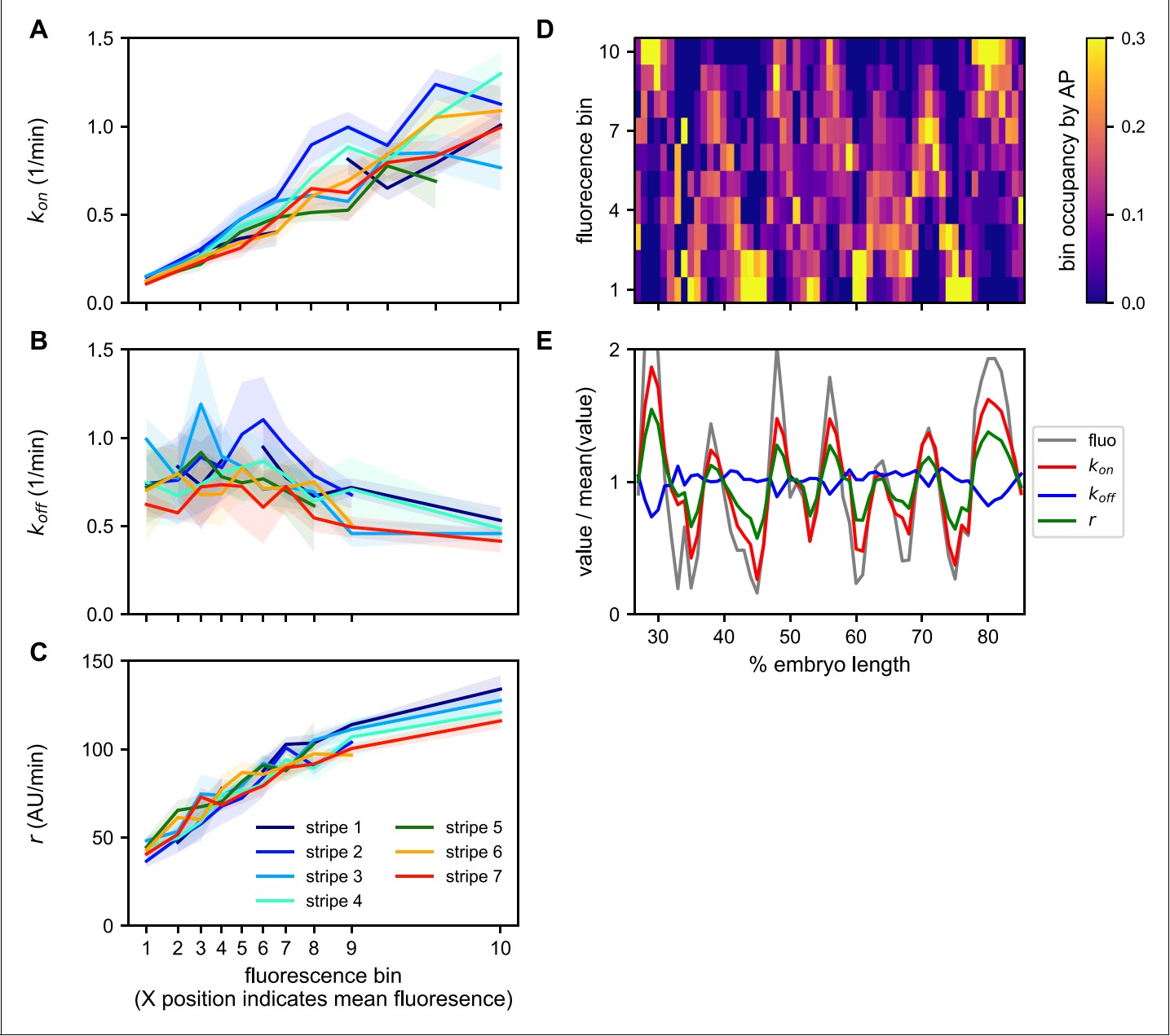

**Figure 8.** A common bursting control mechanism across all *even-skipped* stripes. (A–C) cpHMM inference was carried over nuclei binned according to their average fluorescence value indicating that while (A) $k_{on}$ and (C) $r$ are subject to the same regulation along all stripes, (B) $k_{off}$ remains unchanged. Error bars are calculated by taking the standard deviation across cpHMM inference results for multiple bootstrapped samples of experimental data. (D) Distribution of average nuclear fluorescence values along the AP axis. (E) Mean nuclear fluorescence values for each AP position together with the corresponding averaged and weighted bursting parameters.

that go beyond the control of bursting parameters and that can be as relevant for pattern formation (*Lammers et al., 2020*).

## Stripe movement is driven primarily by expression flow

Just as gene expression patterns are dynamic in time (*Bothma et al., 2014*), they are dynamic in space, resulting in the movement of expression domains throughout the embryo during development (*Jaeger et al., 2004b*; *Keränen et al., 2006*). The anterior movement of eve stripes during

nc14 has been previously described (*Keränen et al., 2006*; *Lim et al., 2018*), and proposed to arise from a combination of nuclear movement (nuclear flow) and movement in the pattern of regulators (expression flow), especially repressors, which are known to shift anteriorly during nc14 as well (*Jaeger et al., 2004b*; *Jaeger et al., 2004a*). While *Keränen et al., 2006* concluded that the relative contributions of these two forces were roughly equal, our data suggest that, especially in the posterior, expression flow dominates the anterior shift of the *eve* stripes.

A typical nucleus in stripe seven moves around one percent of embryo length in the final 25 min of nc14. The stripe, however, moves around five percent of embryo length during that time (see *Figure 7C*). Because we are tracking both the position and activity of individual nuclei, we can visualize expression flow in action. We see nuclei transition from low activity in the anterior interstripe to high activity in the stripe, from high activity in the stripe to low activity off the posterior flank of *eve* expression, and in some cases both.

This effect is most pronounced for the posterior stripes, but is observed for the more anterior stripes as well, although the magnitude of the shift decreases for more anterior stripes (*Figure 7—figure supplement 1*). The difference in the amount of the effect we and Keränen et al. attribute to expression flow is likely an effect of differences in the data used. Because we are looking at instantaneous transcription rates while they looked at accumulated mRNA, there is a considerable temporal lag and integration of the transcriptional activity over the life time of *eve* mRNA in their data, which has the effect of underestimating the extent to which the stripes actually move.

We also note that the extent to which nuclear flow by itself would be expected to shift output patterns measured as a function of position in the embryo is unclear, as it would depend on the extent to which the repositioning of regulators drives movement of nuclei (which it is believed to do [*Blankenship and Wieschaus, 2001*]), and the corresponding effect that nuclear movement has on the positioning of regulators, which remains largely unknown.

One open question relates to the temporal relationship between changes in the position of the repressor array that drives stripe position and the transcriptional output of the stripes. For example, the anterior shift of the stripes of *eve* as well as *fushi tarazu* has been proposed to originate, in part, from cross-repression between these two genes (*Lim et al., 2018*). Recent advances in the simultaneous monitoring of protein concentration and transcriptional output in living embryos should help answer this question in the near future (*Bothma et al., 2018*; *Lammers et al., 2020*).

## Characterizing dynamic patterns demands dynamic measurements

That gene expression is a fundamentally dynamic process is not new information. However, the tools we have had at our disposal to study gene expression so far have tended to emphasize its static features, down to the language we use to describe the transcriptional output of a gene. In textbooks and the scientific literature, *eve* has a gene expression pattern consisting of seven stripes. But, as some earlier work emphasized (*Janssens et al., 2006*), and we have directly visualized here, the transcriptional output of *eve*, rather than a single 'pattern' is a rapidly changing function of time and space: it is dynamic at many time scales and across space and nuclear positions. Indeed, at no point does *eve* approach anything even remotely like a steady state.

We are at the dawn of a new period in the study of transcription, as new experimental techniques and advanced microscopy allow us to monitor transcriptional regulators, observe their behavior at the single-molecule level, and track the transcriptional output of a gene in living, developing animals. We have only barely begun to understand this new data and what it can tell us about biology. While the focus in this paper was on a single gene in a single species, we hope that this and our accompanying work (*Lammers et al., 2020*) will have a broader impact by beginning to establish rigorous frameworks for quantifying, characterizing and visualizing the dynamics of transcription at the single-cell level during development that will be required in the era of live imaging of transcription in development.

## Materials and methods

### Generation of MS2 tagged *eve* BAC

We used bacterial recombineering (*Warming et al., 2005*) to modify a bacterial artificial chromosome (BAC) (*Venken et al., 2006*) containing the *D. melanogaster eve* gene and all of its enhancers

and regulatory elements (BAC CH322-103K22) (*Venken et al., 2009*). We replaced the coding region with an array of 24 MS2 stem loops fused to the *D. melanogaster yellow* gene (*Figure 1B*; *Bothma et al., 2014*) as described below. We inserted our eve::MS2::yellow BAC-based construct in the *D. melanogaster* genome at chromosome 3L through ΦC31 integrase-mediated recombination (see Generation of fly lines), and generated a viable homozygous fly line (w-; +; eve::MS2::yellow) as detailed below.

### Reporter design

In principle the length of the reporter should not limit our ability to estimate burst parameters. However, in practice a reporter construct that is too short will have insufficient signal. Further, one that is too long will increase the dwell time of each RNA polymerase molecule on the gene and, as a result, our cpHMM inference will require too many computational resources. Our choice of reporter construct structure strikes a balance between these two limitations and is ideally suited for inferring bursting parameters in the time range where *eve* resides, as well as for boosting the signal-to-noise ratio. See *Lammers et al., 2020* for a more detailed discussion of reporter length-related tradeoffs.

### Specifics of recombineering

We modified a CHORI BAC CH322-103K22 derived from *Venken et al., 2009*, which contained the entire *eve* locus and a GFP reporter instead of the *eve* coding sequence (CH322-103K22-GFP). We replaced the GFP reporter with MS2::*yellow* (total insert size, 6665 bp) through a two step, scarless, *galK* cassette-mediated bacterial recombineering (*Warming et al., 2005*). Briefly, we transformed our starting CH322-103K22-GFP BAC into *E. coli* recombineering strain SW102. We then electroporated the strain with a *galK* cassette flanked by 50bp-long DNA homology arms homologous to the MS2::*yellow* (6665 bp) reporter. Upon electroporation, we selected transformants on M63 minimal media plates with galactose as a single carbon source. We achieved a correct replacement of GFP sequence by *galK* cassette in the BAC context (CH322-103K22-galK), validated by observing the digestion patterns produced by ApaLI restriction enzyme.

We next purified the CH322-103K22-galK BAC and transformed it into fresh *E. coli* SW102 cells. We electroporated these cells with the purified MS2::*yellow* insert and used M63 minimal media plates with 2-deoxy-galactose to select against bacteria with a functional *galK* gene. We used colony PCR to screen for colonies with a correct MS2::*yellow* insertion (CH322-103K22-MS2) replacing the *galK* cassette. We validated this insertion by observing ApaLI, XhoI, SmaI, and EcoRI restriction digestion patterns and through PCR and Sanger sequencing of the insertion junctions. We transformed our CH322-103K22-MS2 BAC in *E.coli* EPI300 cells to induce high copy numbers and purified it with a Qiagen plasmid Midiprep kit.

### Generation of fly lines

We sent a sample of our purified CH322-103K22-MS2 BAC to Rainbow Transgenic Flies, Inc for injection in *D. melanogaster* embryos bearing a ΦC31 AttP insertion site in chromosome 3L (Bloomington stock #24871; landing site VK00033; cytological location 65B2). We received the flies that resulted from that injection and used a balancer fly line (w-; +; +/TM3sb) to obtain a viable MS2 homozygous line (w-; +; MS2::*yellow*). We used line (yw; His::RFP; MCP::GFP) as the maternal source of Histone-RFP and MCP-GFP (*Garcia et al., 2013*).

### Embryo collection and mounting

Embryo collection and mounting was done as specified in *Garcia and Gregor, 2018*. In short, we set fly crosses between ~30 males (w-; +; eve::MS2::yellow) and ~80 females (yw; His::RFP; MCP::GFP) in a plastic cage capped with a grape juice agar plate. We collected embryos from cages two to ten days old by adding a fresh plate for 30 min and aging for 60 min to target embryos 90 min or younger.

Embryos were mounted on a gas-permeable Lumox Film (Sarstedt - Catalog # 94.6077.317) embedded on a microscope slide hollowed on the center. Then, we coated the hydrophobic side of the Lumox film with heptane glue and let it dry. The film allows for the oxygenation of embryos during the 2–3 hr long imaging sessions while heptane immobilizes them.

We soaked an agar plate with Halocarbon 27 oil, picked embryos with forceps, and laid them down on a 3 × 3 cm piece of paper tissue. We dechorionated embryos by adding 2 drops of bleach diluted in water (5.25%) on the paper tissue and incubating for 1.5 min. We removed bleach with a clean tissue and rinsed with ~4 drops of distilled water. We then placed the tissue paper with dechorionated embryos in water, and picked buoyant embryos with a brush.

We lined ~ 30 apparently healthy embryos on the Lumox film slide and added 2–3 drops of Halocarbon 27 oil to avoid desiccation, and covered the embryos with a cover slip (Corning Cover Glass, No. 1, 18 × 18 mm) for live imaging.

## Imaging and optimization of data collection

Movies of embryonic development were recorded on a Zeiss-800 confocal laser-scanning microscope in two channels, (EGFP: 488 nm; TagRFP: 561 nm). We imaged embryos on a wide field of view, along their anterior-posterior axis, of 1024 × 256 pixels (202.8 x 50.7 µm), encompassing 3–5 stripes per movie. We tuned laser power, scanning parameters, master gain, pinhole size and laser power to optimize signal-to-noise ratio without significant photobleaching and phototoxicity.

For imaging, the following microscope settings were used: 63x oil-objective, scan mode 'frame', pixel size of 0.2 µm, 16 bits per pixel, bidirectional scanning at a speed of 7, line step of 1, laser scanner dwelling per pixel of 1.03 µs, laser scanner averaging of 2, averaging method Mean, averaging mode Line, 488 nm laser power of 30 µW (EGFP), 561 nm laser power of 7.5µW (TagRFP) (both powers were measured with a 10x air-objective), Master Gain in EGFP detector of 550V, Master Gain in TagRFP detector of 650V, Digital Offset in both detectors of 0, Digital Gain in both detectors of 1.0, and a pinhole size of 1 airy unit under the imaging conditions mentioned above (44, 0.7 µm/ section), laser filters EGFP:SP545 and TagRFP:LBF640. This resulted in an imaging time of 633 ms per frame and a full Z-stack of 21 frames in intervals of 0.5 µm every 16.8 s. Following (*Bothma et al., 2014*; *Bothma et al., 2015*; *Bothma et al., 2018*; *Lammers et al., 2020*), the imaging conditions were determined not to affect normal development as reported by the timing of the nuclear cycles in early development. We stopped imaging after 50 min into nuclear cycle 14, and took mid-sagittal and surface pictures of the whole embryo for localization of the recorded field of view along the embryo's AP axis.

## Image processing

We used a Matlab computational pipeline based on *Garcia et al., 2013*; *Lammers et al., 2020* to segment and extract numeric data from our raw movies. Briefly, this software segments and processes the images from the two channels (channel 1: MCP::GFP, channel 2: Histone::RFP) on which we collected our data. For segmentation of channel 1, we used Fiji-Weka Segmentation 3D software; this machine-learning-based method relies on the manual segmentation of a variety of MCP::GFP labeled transcriptional foci in a given 21 frame Z-stack from a single dataset (EVE_D11) to produce a model for the segmentation of all datasets recorded under the same imaging conditions. Next, we segmented and tracked the Histone::RFP-labeled nuclei on channel 2. Subsequently, we assigned MCP::GFP labeled transcriptional foci to their corresponding Histone::RFP-labeled nuclei. Since we collected whole embryo pictures of each of our datasets, we were able to match and locate the recorded fields of view to their right position along the body of their corresponding embryos. Finally, we extracted position and fluorescence values over time of all transcriptional foci to generate data structures ready to use in further analyses.

## Estimation of polymerase transit time

To estimate the transit time of the polymerase along the construct (which is used to determine the persistence of the fluorescence signal from a single transcript at the locus) we first calculated, for each nucleus, the difference in fluorescence signal between adjacent timepoints $D_{n,t} = F_{n,t+1} - F_{n,t}$ where $F_{n,t}$ is the fluorescence signal for nucleus $n$ at time point $t$ and then calculated the Pearson correlation coefficient of the vectors [..., $D_{n,t}$, $D_{n,t+1}$, $D_{n,t+2}$, ..] and [..., $D_{n,t+d}$, $D_{n,t+d+1}$, $D_{n,t+d+2}$, ..] over values of d from 1 to 20 representing time displacements of 20 to 400 s. The minimum correlation occurred at 140 s.

## Compound-state hidden Markov Model

For this work we employed a statistical method that utilizes a compound-state hidden Markov Model to infer bursting parameters from experimental fluorescence traces. The theory and implementation of this method are described in detail in *Lammers et al., 2020*. Briefly, parameters were inferred using a standard version of the Expectation Maximization Algorithm implemented using custom-written scripts in Matlab. Our inference is carried over the full duration of activity of each active nucleus during nuclear cycle 14. Bootstrap sampling was used to estimate the standard error in our parameter estimates. Subsets of 3000 data points were used to generate time-averaged parameter estimates. Inference was not conducted for groups for which fewer than 1000 time points were available.

## Data analysis and figures

All data were analyzed in Python using a Jupyter notebook with custom code to process raw data and generate figures. The Jupyter notebook and all data required to run it is available in *Supplementary file 1* and at (https://github.com/mbeisen/Berrocal_2020; *Berrocal, 2020*; copy archived at swh:1:rev:d983098bd5183f9907d633c425f80b2cb5282a8b).

## Data filtering

We first filtered the raw data to remove data with observations spanning less than 2,000 s, as well as nuclei that were poorly tracked over time defined as nuclei that moved across the movies at an average rate of over 4.5 pixels per minute. This left 430,073 observations from 2959 nuclei.

## Stripe assignment and registration

We used the Gaussian mixture model module of the Python library scikit-learn (*Pedregosa et al., 2011*) to cluster all nuclei time points in each movie in each of a series of overlapping 428 s time windows beginning at 25 min in nc14, specifying the number of components equal to the number of stripes captured in the movie and using the setting *covariance_type='tied'*. We preliminarily assigned nuclei time points to a stripe if they were consistently clustered in that stripe in the relevant time windows. We then pooled all nuclei time points assigned to the same stripe and fit a line to the median x and y positions in the bottom (y < 128) and top (y > 128) halves of the image. We considered the slope of this line to represent the orientation of the stripe to the image x-axis. We then went back to each time window and fit the nuclei assigned to the stripe with a line with the previously computed slope fixed. This produced an association of time with stripe position, from which we derived a linear model that describes the position of each stripe in each movie at every time point.

We assigned all nuclei time points (not just bursting ones) to stripes by identifying the stripe whose predicted position at the relevant time was closest (along the x-axis) to the nucleus being analyzed, and assigned a nucleus to the most common stripe assignment for its individual time points. We then corrected the reorientation of the stripe at each time point to be perpendicular to the image x-axis (to enable projection along the AP axis) by setting its new image x-axis position to be the x position of the stripe in the middle of the y-axis (y = 128) plus the offset of the nucleus to the unoriented stripe along the x-axis. Finally, we used the positions of the anterior and posterior poles of the embryo to map image x coordinates to AP position. We then adjusted the AP position of each stripe in each movie such that the center of the stripe at 35 min in nc14 had the same AP position.

## Acknowledgements

This work was supported by an HHMI Investigator award to MBE. HGG was supported by the Burroughs Wellcome Fund Career Award at the Scientific Interface, the Sloan Research Foundation, the Human Frontiers Science Program, the Searle Scholars Program, the Shurl and Kay Curci Foundation, the Hellman Foundation, the NIH Director's New Innovator Award (DP2 OD024541-01), and an NSF CAREER Award (1652236). NL was supported by NIH Genomics and Computational Biology training grant 5T32HG000047-18. AB was supported by a Doctoral Fellowship from The University of California Institute for Mexico and the United States (UC MEXUS) and CONACyT.

## Additional information

### Competing interests
Michael B Eisen: Editor-in-Chief, *eLife*. The other authors declare that no competing interests exist.

### Funding

| Funder | Grant reference number | Author |
|---|---|---|
| Howard Hughes Medical Institute | Investigator Award | Michael B Eisen |
| National Science Foundation | 1652236 | Hernan G Garcia |
| National Institutes of Health | DP2-OD024541-01 | Hernan G Garcia |
| National Institutes of Health | 5T32HG000047-18 | Nicholas C Lammers |
| University of California Institute for Mexico and the United States | | Augusto Berrocal |
| Burroughs Wellcome Fund | | Hernan G Garcia |
| Alfred P. Sloan Foundation | | Hernan G Garcia |
| Human Frontier Science Program | | Hernan G Garcia |
| Searle Scholars Program | | Hernan G Garcia |
| Shurl and Kay Curci Foundation | | Hernan G Garcia |
| Hellman Foundation | | Hernan G Garcia |

The funders had no role in study design, data collection and interpretation, or the decision to submit the work for publication.

### Author contributions
Augusto Berrocal, Data curation, Methodology; Nicholas C Lammers, Software, Formal analysis, Visualization, Methodology; Hernan G Garcia, Conceptualization, Supervision, Funding acquisition, Methodology, Writing - original draft, Project administration, Writing - review and editing; Michael B Eisen, Conceptualization, Resources, Data curation, Software, Formal analysis, Supervision, Funding acquisition, Visualization, Writing - original draft, Project administration, Writing - review and editing

### Author ORCIDs
Augusto Berrocal (iD) https://orcid.org/0000-0002-6506-9071
Nicholas C Lammers (iD) https://orcid.org/0000-0001-6832-6152
Hernan G Garcia (iD) https://orcid.org/0000-0002-5212-3649
Michael B Eisen (iD) https://orcid.org/0000-0002-7528-738X

### Decision letter and Author response
Decision letter https://doi.org/10.7554/eLife.61635.sa1
Author response https://doi.org/10.7554/eLife.61635.sa2

## Additional files

### Supplementary files
• Supplementary file 1. Data and code.
• Transparent reporting form

## Data availability

All of the raw and processed data described in this paper are available on Data Dryad at https://doi.org/10.6078/D1XX33 and computational notebooks with necessary data to regenerate analyses and figures is available in Supplementary file 1 and at https://github.com/mbeisen/Berrocal_2020 (copy archived at https://archive.softwareheritage.org/swh:1:rev:d983098bd5183f9907d633c425f80b2cb5282a8b/).

The following dataset was generated:

| Author(s) | Year | Dataset title | Dataset URL | Database and Identifier |
|---|---|---|---|---|
| Eisen MB | 2018 | Kinetic sculpting of the seven stripes of the *Drosophila* even-skipped gene | https://doi.org/10.6078/D1XX33 | Dryad Digital Repository, 10.6078/D1XX33 |

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
