## [Decision Letter]

**Acceptance summary:**

The authors used live imaging to visualize the transcriptional dynamics of the *Drosophila* even-skipped gene at single-cell resolution during stripe formation, and developed tools to characterize and visualize how the role of transcriptional bursting influences stripe formation. By tracking the position and activity of individual nuclei they find stripe movement is driven by the exchange of bursting nuclei from the posterior to anterior stripe flanks.

**Decision letter after peer review:**

[Editors’ note: the authors submitted for reconsideration following the decision after peer review. What follows is the decision letter after the first round of review.]

Thank you for submitting your article "Kinetic sculpting of the seven stripes of the *Drosophila even-skipped* gene" for consideration by *eLife*. Your article has been reviewed by three peer reviewers, one of whom is a member of our Board of Reviewing Editors, and the evaluation has been overseen by Naama Barkai as the Senior Editor. The reviewers have opted to remain anonymous.

The reviewers have discussed the reviews with one another and the Reviewing Editor has drafted this decision to help you prepare a revised submission.

All reviewers felt the work has merit but have some concerns that are expressed below.

Essential revisions:

You will need to address the question about whether the reporter genes disrupt or recapitulate the endogenous genes. It seems some experimentation might be done here to resolve this-perhaps a FISH experiment. Ideally the endogenous gene could be tagged. Data may exist for this point that is not in the manuscript.

The second reviewer has issues with the analysis at several points. These need to be addressed.

The third reviewer has points that need to be addressed that mainly involve further clarifications in the text and focusing the Discussion.

Reviewer #1:

This is a companion manuscript to Lammers, et al. It increases the number of stripes analysed for the even skipped gene, assessed using the method of modeling using a modified HMM analysis to provide a memory of the transcriptional bursting, and to include the duration of this bursting in defining the sharp boundaries of the stripes. The technical advance in this work is to insert the entire even skipped set of enhancers and regulatory elements on a BAC so that the ontogeny of all seven stripes can be observed at high time resolution (20 second intervals).

This is an elegant piece of work, although with some of the same caveats that are relevant for the Lammers manuscript. One concern is whether the embryos would develop past gastrulation, given the light dosage (Sedat has made a big point about this). It would be useful to know this, although it may not influence conclusions about the early events. As with the Lammers manuscript, the observations are by their nature descriptive, but provide a basis for evaluations of various mechanisms; these are spelled out well in the Discussion, with appropriate cautionary caveats. One of the more interesting observations is that the stripes seem to be similarly regulated in kinetics and duration of the transcription despite the variety of factors regulating them. This commonality of mechanism requires some new thinking about the role of these various elements.

Some quibbles about the writing style: the concept of a "transcriptional desert" seems a bit melodramatic (as does the repetitive "stripe is a stripe..." line) and obscures the mechanistic implications. Presumably the "desert" is just the closing of the transcriptional window, as described more precisely in the accompanying manuscript. To further impinge on the author's expository flair here, the concept of "sculpting" is too anthropomorphic, as one needs a sculptor to sculpt and this is conceptually confusing since these events are stochastically driven. Is the sculptor a facilitator of "intelligent design"? Do you want to go there?

Some of my concerns about the Lammers manuscript can be reiterated briefly. The inserted *eve* reporter system is now sucking up factors and presumably competing with the endogenous gene possibly causing a "transcriptional haploinsufficiency" (my own terminological flair in this case) wherein the effective local concentrations are now half what they should be, and this may have the effect of changing the bursting and duration kinetics. Furthermore, the local environment of the endogenous gene may contain more regulatory chromatin not dreamt of in your philosophy, that would rule the expression of *eve* with a firmer hand (using the anthropomorphizing theme). Hence the assertion that the inserted gene "is phenotypically neutral" seems glib and may not be technically correct, at least until tested. Assuming that the function of *eve* is not compromised by some of the modifications used for the reporter gene, why not tag the endogenous gene? (I may be unaware of literature that relates to this experiment, but I don't think that complementation in a mutant strain reproduces the robustness of the endogenous gene).

All told, I feel that these considerations could be dealt with in the Discussion.

Reviewer #2:

In this work by Berrocal et al., the authors provide a quantitative treatment of the enduring developmental problem of how the combinatorial inputs into an enhancer drive spatial patterns in the blastoderm. Work in recent years indicates these developmental genes are also stochastic and exhibit bursting transcription, making the emergence of these ordered patterns even more difficult to explain. This manuscript is decidedly observational yet also highly quantitative. No molecular mechanisms are elucidated, but the underlying spatiotemporal transcription phenomena are characterized with experimental rigor. The authors seem to believe they have uncovered some universal behaviors but they stop short of making any strong claims, making it hard for this reviewer to come away with a digestible conclusion. Overall, I found the paper to be scholarly and well-written. I think it could be appropriate for *eLife* with some revisions.

1) I found the paper to be somewhat off balance in the presentation. Some of the concepts and data analyses are covered in a cursory manner, even for specialists, but the Discussion goes on at length. For example, I am confused by Figure 4A. Why does the autocorrelation go negative? What does the autocorrelation of the first derivative even mean? Is this experimental data?

2) There is a tremendous amount of live-cell data, and I appreciate the scale of that analysis effort. The two major inference problems in this data analysis are the following: 1) promoter activity must be inferred through direct analysis of the transcription time traces, and 2) burst kinetics have to be assigned to a particular stripe. I would like to see some straightforward analysis or metric of how good these inferences are. Error bars or confidence analysis is noticeably absent throughout the manuscript.

3) Related to this second inference, I found the exceptional behavior of the 5 – 6 interstripe to be important. Is this looser regulation also observed by smFISH? Such an analysis would also indicate how well the stripe assignment works in live blastoderms. Is there any biological functionality to the 5-6 interstripe bursting?

4) In Figure 8, is there some sort of reduced parameterization which captures the essential behavior, rather than an empirical tabulation of on and off rates? The results in Figure 10 argue for such a reduced parameterization, and the comparison is made to the Zoller paper in the Discussion, but I don't see where they actually test a reduced model for fitting the data.

5) Related to this last point, when I look at Figure 8 , I see a lot of variability at the low ON rates. This variability could result in an apparent trend which is artefactual. At some point, there is a noise floor where the mHMM might be picking up single infrequent bursts that could therefore be spurious. I find the treatment of this key point unsatisfactory, both because there is not an in depth statistical analysis, and also because the authors use sentences such as "To investigate whether this result is an artifact of the mHMM, we implemented an orthogonal method that uses integer programming to infer promoter states from traces based on a direct fit of 140s fluorescence pulses." This one-sentence description is obtuse, even for specialists. I think there needs to be greater effort here.

Reviewer #3:

In this work, Berrocal and Lammers et al. propose a conceptual, theoretical and computational framework for dissecting gap genes pattern formation in space and time in early *Drosophila* embryos. The framework is showcased in the analysis of the dynamics of the even-skipped gene, monitored at the single-cell and high temporal resolution.

This is an ambitious and much-needed work targeting the transcription dynamics in mid and late nuclear cycle 14. Here, the MS2-GFP system is employed to simultaneously monitor the transcription dynamics of the even-skipped gene in seven stripes, each controlled by the independent activity of a specific enhancer: whereas stripe 1, 2 and 5 are each controlled by a different enhancer, stripes 3 and 7 and stripes 4 and 6 are controlled by a common enhancer. This provides the basis for the analysis of the transcription kinetics, as demonstrated here with the memory-adjusted Hidden Markov Model. In this last long nuclear cycle before gastrulation, the transcription factor gradients are transient. Thus, traditional analyses are hindered due to the movement of both the nuclei and transcription factors distributions. From the videos, the paper presents a protocol to classify and track the stripes position over time, and attributing nuclei to the corresponding stripes.

The main conclusion is that despite being created by the activity of independent enhancers, even-skipped stripes emerged from globally similar kinetic phenomena over time which combine an increase of transcription burst frequency and duration within the stripes and a progressive reduction of bursting between stripes. Interestingly, the dynamics and the timing of stripe formation appears quite different among stripes but does not correlate with the enhancer regulating the stripe formation (for example, the kinetics features of stripes 4 and 6 and of stripe 3 and 7 are not more similar to each other than they are to the five other stripes). Thus, despite being created by the independent activity of different enhancers, even-skipped stripes exhibit similar kinetic phenomena over time. This is a very interesting observation, suggesting the co-evolution of enhancers to ensure the robustness of the downstream processes.

1) The five even-skipped enhancers have been identified using transgenic reporters and mostly experiments on fixed embryos (ISH or FISH) that did not capture the fast dynamics uncovered with the MS2 system and live imaging. It is thus not clear how these five enhancers might recapitulate the different dynamics observed here for the seven stripes with the MS2 system. Although it is beyond the scope of this manuscript, it is important to mention in the manuscript potential discrepancies between the two experimental approaches.

2) It has been shown recently that the MS2 cassette itself might influence the regulation of the target promoter (Lucas et al., 2018, bioRxiv). It is thus important to mention in the manuscript the possibility that the common regulatory properties observed for the seven stripes could be driven by the MS2 cassette. Could the authors confront the dynamics of expression of the seven stripes that they uncover with the MS2 system from FISH data on precisely staged cycle 14 embryos?

3) As mentioned by the authors, the length of the construct is very long (6, 5 kb) and carry the MS2 cassette in 5' of the transcribed sequence. This was done on purpose to increase the fluorescent signal but also impairs the sensitivity in detection fluctuations. A good example of this is shown in Figure 4C where a small activity of the promoter provides a very strong fluorescent signal. Can the authors comment on this and indicate the range of kinetics parameters that this construct allows to capture?

4) Most of the works' results and conclusions are based on the temporal k_on_ and k_off_, which can be extracted directly for each nucleus. This is very likely to be confused with k_on_ and k_off_ inferred from the mHMM approach. Please discuss how the temporal k_on_ and k_off_ are calculated when they are first mentioned (we only find out about this until the very end of the manuscript) and use a different annotation (or markers) to avoid confusion with mHMM's.

5) Given the ON and OFF rate changing over time as pointed out above, one should question the validity of the mHMM here. Can the authors clarify on this point?

6) In the Materials and methods section, how is the time window for each nucleus selected for mHMM inference? Is it the time window when the nucleus is in the temporal stripe region?

7) Given the movement of the transcription stripes over time, protein stripes which are translated over a range of AP position should be wider as mRNAs stripes. Can the authors show how the integrated transcription stripes look like over time, or at the end of the cycles?

8) The Discussion of the manuscript is too long and would benefit of being more focused.

---

## [Author Response]

[Editors’ note: the authors resubmitted a revised version of the paper for consideration. What follows is the authors’ response to the first round of review.]

Reviewer #1:[…] This is an elegant piece of work, although with some of the same caveats that are relevant for the Lammers manuscript. One concern is whether the embryos would develop past gastrulation, given the light dosage (Sedat has made a big point about this). It would be useful to know this, although it may not influence conclusions about the early events.

Author response: The imaging conditions are identical to those used repeatedly in various previous experiments (e.g., Bothma et al., 2014, Bothma et al., 2015; Bothma et al., 2018; Lammers et al., 2020). Here, the light dose was determined to be appropriate so as not to affect development as reported by the timing of the nuclear cycles in early development. We have now made this point explicitly in the Materials and methods section under “Imaging and Optimization of Data Collection”.

As with the Lammers manuscript, the observations are by their nature descriptive, but provide a basis for evaluations of various mechanisms; these are spelled out well in the Discussion, with appropriate cautionary caveats. One of the more interesting observations is that the stripes seem to be similarly regulated in kinetics and duration of the transcription despite the variety of factors regulating them. This commonality of mechanism requires some new thinking about the role of these various elements.Some quibbles about the writing style: the concept of a "transcriptional desert" seems a bit melodramatic (as does the repetitive "stripe is a stripe..." line) and obscures the mechanistic implications. Presumably the "desert" is just the closing of the transcriptional window, as described more precisely in the accompanying manuscript. To further impinge on the author's expository flair here, the concept of "sculpting" is too anthropomorphic, as one needs a sculptor to sculpt and this is conceptually confusing since these events are stochastically driven. Is the sculptor a facilitator of "intelligent design"? Do you want to go there?

While we were simply trying to add a little rhetorical color to the manuscript, we accept the reviewer’s admonition that they are distracting and perhaps misleading and have adjusted our language accordingly. For example, in the new version of the manuscript, we have eliminated all reference to transcriptional deserts. The one area where we disagree is with the term “sculpting”. While we recognize that it has some anthropomorphic character, here it was meant as a metaphor to emphasize that, by adjusting the kinetics of bursting in individual nuclei, a complex pattern emerges from an initially unfinished and unfeatured mass. It is not a hill on which we would die, but we hope the reviewers will allow us this flourish.

Some of my concerns about the Lammers manuscript can be reiterated briefly. The inserted eve reporter system is now sucking up factors and presumably competing with the endogenous gene possibly causing a "transcriptional haploinsufficiency" (my own terminological flair in this case) wherein the effective local concentrations are now half what they should be, and this may have the effect of changing the bursting and duration kinetics. Furthermore, the local environment of the endogenous gene may contain more regulatory chromatin not dreamt of in your philosophy, that would rule the expression of eve with a firmer hand (using the anthropomorphizing theme). Hence the assertion that the inserted gene "is phenotypically neutral" seems glib and may not be technically correct, at least until tested. Assuming that the function of eve is not compromised by some of the modifications used for the reporter gene, why not tag the endogenous gene? (I may be unaware of literature that relates to this experiment, but I don't think that complementation in a mutant strain reproduces the robustness of the endogenous gene).

There are two separate concerns being addressed here. First, that the regulation of a BAC may not be the same as the endogenous locus, and second that the BAC itself might be disrupting its own regulation by adding an extra locus to the system.

In our accompanying paper (Supplementary Figure 9 in Lammers et al., 2020) we have actually shown that, given reasonable values for the *eve* mRNA degradation rate, the profile resulting from our BAC is quantitatively comparable to that of the endogenous *eve* locus as measured by mRNA FISH. We now mention this control in the section “Live imaging of *eve* expression”. Further, while CRISPR insertions of MS2 (or PP7) to the *eve* locus exist, these either don’t rescue (Chen et al. , Nat Genet, 2018) or are located on the 3’ end of the gene (Lim et al., 2018) such that the fluorescence signal is significantly reduced (as described in Garcia et al., 2013 and Ferraro et al., WIREs Dev Biol, 2016).

The second concern, that the addition of an extra copy of a locus might affect gene regulation by diluting the concentration of regulators is interesting, but we think this phenomenon is highly unlikely to have a measurable effect here. First, all of the well-characterized regulators of *eve* (Bcd, Hb, Gt, Kr, Kni, Slp1) have many targets throughout the genome, and bind to thousands of sites in the blastoderm (see Li et al., PLoS Biology, 2008). Thus the addition or removal of a single locus to the system would change the effective concentration of the factors by one percent at most – far less than the natural nucleus to nucleus noise in transcription factor levels (see Gregor et al., Cell, 2007) or variation in the size of nuclei within an embryo. Indeed it would be truly remarkable to have a regulatory system that could respond to such a small change in factor concentration. Furthermore, we expect selection to strongly disfavor such sensitivity, as it runs counter to the need to maintain robustness in gene expression outputs during development. We hope the reviewer will agree with us that this “transcriptional haploinsufficiency” is probably a small effect and that it is not worth mentioning explicitly in the manuscript so as not to dilute its message.

All told, I feel that these considerations could be dealt with in the Discussion.

We have addressed the reviewer’s concerns throughout the text. In addition, the Discussion section has been significantly modified to better focus on the insights about the molecular control of transcriptional bursting resulting from our work.

Reviewer #2:In this work by Berrocal et al., the authors provide a quantitative treatment of the enduring developmental problem of how the combinatorial inputs into an enhancer drive spatial patterns in the blastoderm. Work in recent years indicates these developmental genes are also stochastic and exhibit bursting transcription, making the emergence of these ordered patterns even more difficult to explain. This manuscript is decidedly observational yet also highly quantitative. No molecular mechanisms are elucidated, but the underlying spatiotemporal transcription phenomena are characterized with experimental rigor. The authors seem to believe they have uncovered some universal behaviors but they stop short of making any strong claims, making it hard for this reviewer to come away with a digestible conclusion. Overall, I found the paper to be scholarly and well-written. I think it could be appropriate for eLife with some revisions.

We thank the reviewer for carefully assessing our manuscript. We appreciate the feedback, both specific and general, and have tried to address it in the revised manuscript. Specifically to the reviewer's point, we have revised the Discussion section to more focus on the potential underpinnings of the control mechanisms for bursting uncovered by our work. Having said that, we have also added a call for new experiments that utilize the methods developed here to further probe into how transcription factors actually realize the control of bursting.

1) I found the paper to be somewhat off balance in the presentation. Some of the concepts and data analyses are covered in a cursory manner, even for specialists, but the Discussion goes on at length.

As mentioned above we have significantly edited the Discussion section to make it more focused. In addition, we have expanded the description of how our analyses are done, and removed superfluous analyses that might have added more confusion than clarification.

For example, I am confused by Figure 4A. Why does the autocorrelation go negative? What does the autocorrelation of the first derivative even mean? Is this experimental data?

We apologize for our lack of clarity. We have now revised the description of the autocorrelation approach in section “Modeling and inference of promoter state” and in the Materials and methods section under “Estimation of polymerase transit time”. The idea is that every loaded polymerase induces a fluorescence signal for essentially the entire time it is transiting the locus: a pulse of fluorescence of width equal to the transit time. Thus the increase in signal associated with a polymerase loaded at time t should be coupled to a decrease in signal at t + transit time. We reveal this lag by looking at the autocorrelation of the vector of the timepoint by timepoint change in fluorescence signal, which has a minimum at 140s (we did 100 bootstraps over randomly chosen sets of 80% of all nuclei, which give almost identical results that are now shown in Figure 4A).

2) There is a tremendous amount of live-cell data, and I appreciate the scale of that analysis effort. The two major inference problems in this data analysis are the following: 1) promoter activity must be inferred through direct analysis of the transcription time traces, and 2) burst kinetics have to be assigned to a particular stripe. I would like to see some straightforward analysis or metric of how good these inferences are. Error bars or confidence analysis is noticeably absent throughout the manuscript.

The challenge here is that we lack the means of knowing the actual state of the promoter (this is something we are working on but it is a long-term challenge that is beyond the scope of this work). In Lammers et al., 2020, we describe the effectiveness of our inference to recover model parameters from simulated data. However, this is under the still unproven assumption that the base model implicit in the cpHMM is correct. All of these points are made explicitly in the Lammers et al. work, but we are happy to repeat them in this manuscript if the reviewer deems it necessary.

We do apologize for the lack of error bars throughout the paper. Where possible, we have now added error bars generated by, for example, bootstrapping over the nuclei in our data set. Specifically, we have added error bars to the single-cell fluorescence traces shown in Figure 3A, to the inference of the RNAP dwell time stemming from the autocorrelation analysis (Figure 4A), and to our inferred bursting parameters (Figures 8A-C).

3) Related to this second inference, I found the exceptional behavior of the 5 – 6 interstripe to be important. Is this looser regulation also observed by smFISH? Such an analysis would also indicate how well the stripe assignment works in live blastoderms. Is there any biological functionality to the 5-6 interstripe bursting?

This is indeed a very interesting question (though outside of the scope of the current work). Given the fast pace of the pattern as reported by our data and the low temporal resolution attainable by smFISH, however, we doubt that the latter technique can shed light on the detailed behavior of the 5-6 interstripe.

4) In Figure 8, is there some sort of reduced parameterization which captures the essential behavior, rather than an empirical tabulation of on and off rates? The results in Figure 10 argue for such a reduced parameterization, and the comparison is made to the Zoller paper in the Discussion, but I don't see where they actually test a reduced model for fitting the data.

Note that, in Figure 8, we have now performed inference by binning our nuclei according to their average fluorescence values (which reports on the average transcriptional activity of each nucleus) rather than grouping them by their position along the embryo. In doing so, the correlations between *k**_on_* and *k**_off_* vanished (Figure 8A and B). We believe that this is due, in part, to the large nucleus-to-nucleus variation in average fluorescence values at a given AP position (Figure 8D). Instead, when binning by fluorescence, we find that *k**_on_* increases with fluorescence while *k**_off_* remains constant. Further, we find that *r* also increases with fluorescence (Figure 8C).

Based on these results, we could have performed an inference with a model where *k**_on_* and *r* are coupled. We hope the reviewer will agree with us that such exploration of a model that goes beyond the widespread two-state model of bursting calls for a solid theoretical footing that goes beyond the scope of the current work, but that we are very excited to pursue in the future.

5) Related to this last point, when I look at Figure 8 , I see a lot of variability at the low ON rates. This variability could result in an apparent trend which is artefactual. At some point, there is a noise floor where the mHMM might be picking up single infrequent bursts that could therefore be spurious. I find the treatment of this key point unsatisfactory, both because there is not an in depth statistical analysis, and also because the authors use sentences such as "To investigate whether this result is an artifact of the mHMM, we implemented an orthogonal method that uses integer programming to infer promoter states from traces based on a direct fit of 140s fluorescence pulses." This one-sentence description is obtuse, even for specialists. I think there needs to be greater effort here.

We agree that there are many potential artifacts clouding out inference. As a result, we have removed any analyses that rely on inferences in regimes such as the low ON-rate one described by the reviewer. Further, we have performed a systematic analysis of the limitations of the cpHMM method using synthetic data in Lammers et al.  which we would be happy to reproduce here if the reviewer deems it necessary.

Reviewer #3:[…] 1) The five even-skipped enhancers have been identified using transgenic reporters and mostly experiments on fixed embryos (ISH or FISH) that did not capture the fast dynamics uncovered with the MS2 system and live imaging. It is thus not clear how these five enhancers might recapitulate the different dynamics observed here for the seven stripes with the MS2 system. Although it is beyond the scope of this manuscript, it is important to mention in the manuscript potential discrepancies between the two experimental approaches.

In our accompanying paper (Supplementary Figure 9 in Lammers et al., 2020) we have actually shown that, given reasonable values for the *eve* mRNA degradation rate, the profile resulting from our BAC is quantitatively comparable to that of the endogenous *eve* locus as measured by mRNA FISH. We now mention this control in the section “Live imaging of *eve* expression”.

2) It has been shown recently that the MS2 cassette itself might influence the regulation of the target promoter (Lucas et al., 2018, bioRxiv). It is thus important to mention in the manuscript the possibility that the common regulatory properties observed for the seven stripes could be driven by the MS2 cassette. Could the authors confront the dynamics of expression of the seven stripes that they uncover with the MS2 system from FISH data on precisely staged cycle 14 embryos?

This is a very important point indeed as Lucas et al. have shown that the presence of Zelda sites in the MS2 loops alters the expression dynamics of the *hunchback* P2 enhancer and promoter. However, given the direct comparison between the BAC and the endogenous *eve* patterns (Supplementary Figure 9 in Lammers et al., 2020) mentioned in the previous point, we believe that, unlike in the case reported by Lucas et al., the MS2 stem loops do not significantly affect *eve* regulation. The reasons for this remain unclear, but could be related to different amounts of Zelda present at both loci. For example, if Zelda binding is already saturated at *eve*, we would not expect the presence of more Zelda sites in the MS2 loops to have a significant effect. Further, the distance between the loops and the enhancers might also make a difference. While the *hunchback* P2 promoter and enhancer are essentially one contiguous unit on the DNA, the *eve* enhancers are separated from the promoter and distributed throughout the locus.

3) As mentioned by the authors, the length of the construct is very long (6, 5 kb) and carry the MS2 cassette in 5' of the transcribed sequence. This was done on purpose to increase the fluorescent signal but also impairs the sensitivity in detection fluctuations. A good example of this is shown in Figure 4C where a small activity of the promoter provides a very strong fluorescent signal. Can the authors comment on this and indicate the range of kinetics parameters that this construct allows to capture?

This is a very important point and one that we explore in detail in Lammers et al., 2020. We have now referred the reader to this work when we introduce the reporter construct in the section titled “Reporter Design” within the Materials and methods. In principle the length of the reporter should not limit our ability to estimate burst parameters; however, in practice a construct that is too short will have insufficient signal and one that is too long will require too many computational resources. Our choice of reporter construct structure strikes a balance between these two limitations and is ideally suited for inferring bursting parameters in the time range where *eve* resides, as well as for boosting the signal-to-noise ratio.

4) Most of the works' results and conclusions are based on the temporal k_on_ and k_off_, which can be extracted directly for each nucleus. This is very likely to be confused with k_on_ and k_off_ inferred from the mHMM approach. Please discuss how the temporal k_on_ and k_off_ are calculated when they are first mentioned (we only find out about this until the very end of the manuscript) and use a different annotation (or markers) to avoid confusion with mHMM's.

We agree with the reviewer that the treatment of this different set of kinetic

parameters could result in potential confusion. As a result, we have decided to focus on the *k**_on_*, *k**_off_* and *r* parameters inferred by the HMM model and have removed our analyses based on inferring these parameters from single-nucleus traces.

5) Given the ON and OFF rate changing over time as pointed out above, one should question the validity of the mHMM here. Can the authors clarify on this point?

The temporal variation in bursting parameters is indeed an issue. In light of this, we have decided not to focus on the temporal variation in the rates, except with respect to stripe movements, and focus our inference on a period of time when expression seems to be stable.

6) In the Materials and methods section, how is the time window for each nucleus selected for mHMM inference? Is it the time window when the nucleus is in the temporal stripe region?

Our inference is carried over the full duration of activity of each active nucleus during nuclear cycle 14. We have clarified this point in the Materials and methods, section “Compound-state Hidden Markov Model”.

Further, to simplify our paper, we have decided to exclude the inference of the temporal dependence of bursting parameters in favor of focusing on the commonalities of bursting parameters between different stripes (Figure 8).

7) Given the movement of the transcription stripes over time, protein stripes which are translated over a range of AP position should be wider as mRNAs stripes. Can the authors show how the integrated transcription stripes look like over time, or at the end of the cycles?

Such analysis is possible, but relies on assuming a value for the mRNA and protein degradation rates. While we have engaged in such exercise in the past (Bothma et al., 2014), we worry that the analysis proposed by the reviewer would dilute the message of the work.

8) The Discussion of the manuscript is too long and would benefit of being more focused.

We agree with the reviewer and have significantly shortened and focussed the Discussion section.